# Emerging Approaches for Mitigating Biofilm-Formation-Associated Infections in Farm, Wild, and Companion Animals

**DOI:** 10.3390/pathogens13040320

**Published:** 2024-04-13

**Authors:** Daniela Araújo, Ana Rita Silva, Rúben Fernandes, Patrícia Serra, Maria Margarida Barros, Ana Maria Campos, Ricardo Oliveira, Sónia Silva, Carina Almeida, Joana Castro

**Affiliations:** 1INIAV—National Institute for Agrarian and Veterinarian Research, Rua dos Lagidos, 4485-655 Vila do Conde, Portugal; 10221259@ess.ipp.pt (A.R.S.); rubenfernandes@ipvc.pt (R.F.); pmserra02@gmail.com (P.S.); margarida.barros@iniav.com (M.M.B.); anamaria.campos@iniav.pt (A.M.C.); ricardo.oliveira@iniav.pt (R.O.); soniasilva@ceb.uminho.pt (S.S.); carina.almeida@iniav.pt (C.A.); 2CEB—Centre of Biological Engineering Campus de Gualtar, University of Minho, 4710-057 Braga, Portugal; 3LABBELS—Associate Laboratory, 4710-057 Braga, Portugal; 4CECAV—Veterinary and Animal Research Centre, University of Trás-os-Montes and Alto Douro, 5000-801 Vila Real, Portugal; 5LEPABE—Laboratory for Process Engineering, Environment, Biotechnology and Energy, Faculty of Engineering, University of Porto, Rua Dr. Roberto Frias, 4200-465 Porto, Portugal; 6AliCE—Associate Laboratory in Chemical Engineering, Faculty of Engineering, University of Porto, Rua Dr. Roberto Frias, 4200-465 Porto, Portugal

**Keywords:** animals, infections, biofilms, antimicrobial resistance, novel biofilm treatments

## Abstract

The importance of addressing the problem of biofilms in farm, wild, and companion animals lies in their pervasive impact on animal health and welfare. Biofilms, as resilient communities of microorganisms, pose a persistent challenge in causing infections and complicating treatment strategies. Recognizing and understanding the importance of mitigating biofilm formation is critical to ensuring the welfare of animals in a variety of settings, from farms to the wild and companion animals. Effectively addressing this issue not only improves the overall health of individual animals, but also contributes to the broader goals of sustainable agriculture, wildlife conservation, and responsible pet ownership. This review examines the current understanding of biofilm formation in animal diseases and elucidates the complex processes involved. Recognizing the limitations of traditional antibiotic treatments, mechanisms of resistance associated with biofilms are explored. The focus is on alternative therapeutic strategies to control biofilm, with illuminating case studies providing valuable context and practical insights. In conclusion, the review highlights the importance of exploring emerging approaches to mitigate biofilm formation in animals. It consolidates existing knowledge, highlights gaps in understanding, and encourages further research to address this critical facet of animal health. The comprehensive perspective provided by this review serves as a foundation for future investigations and interventions to improve the management of biofilm-associated infections in diverse animal populations.

## 1. Introduction

The prevalence of biofilm-associated infections in farm, wild, and companion animals has become a major concern in veterinary medicine. As it has been well documented, biofilms, complex communities of microorganisms embedded in a self-produced extracellular matrix, contribute to persistent and difficult-to-treat infections [1,2].

Biofilms provide a protective environment for microorganisms, increasing their resistance or tolerance to antimicrobial agents and the host immune response [2]. This resistance often leads to chronic infections, in farm, companion, and wild animals, affecting their overall health and welfare. These persistent infections may cause discomfort, pain, and reduced reproductive success, impacting the quality of life for individual animals [3].

In farm animals, biofilm formation poses a considerable economic threat to the agricultural industry. Chronic infections result in reduced productivity, compromised meat and milk quality, and increased veterinary costs [4]. Addressing biofilm-related challenges is crucial for maintaining sustainable and profitable farming practices. Furthermore, the environmental consequences of biofilm-related challenges are notable. In farm settings, excess use of antimicrobials to combat biofilms can contribute to antibiotic resistance and environmental pollution [5]. Understanding and mitigating these impacts is essential for promoting sustainable agricultural and environmental practices. The impact of biofilms on companion animals can be significant and has implications for both the health of the animals and the challenges faced by veterinarians in diagnoses and treatment. Biofilm-associated infections are resistant to conventional antibiotic therapy [1,3]. The protective matrix of the biofilm limits the effectiveness of antibiotics, making it difficult to completely eradicate the infection. This can result in prolonged and recurrent treatment regimens [6]. Biofilms can also affect the health of wildlife populations, influencing species abundance and diversity. In the context of wildlife conservation, understanding and managing biofilm-associated infections are critical for maintaining the ecological balance and biodiversity within ecosystems [7].

Biofilm-forming microorganisms in animals may serve as reservoirs for potential zoonotic pathogens, posing risks to human health [3]. Studying and managing biofilms in farm, companion, and wild animals contribute to preventing the transmission of infectious diseases between animals and humans. Understanding the mechanisms of biofilm formation and exploring innovative strategies to mitigate its impact is essential to ensure animal welfare and sustainable food production. In this comprehensive review, we aim to explore new approaches and advances in biofilm management specific to livestock, wildlife, and companion animals. By synthesizing current knowledge, we aim to shed light on novel interventions that hold promise for the prevention, control, and treatment of biofilm-associated infections in different animal settings. The integration of cutting-edge research and practical applications will not only benefit animal health but also address the wider impacts of biofilm-related challenges on the agricultural and environmental landscape.

## 2. Biofilm Formation

Biofilms are defined as a population of microbial cells that are permanently bound together on a biotic or abiotic surface by an extracellular polymeric substance (EPS) matrix [8,9]. Biofilms can cause severe and persistent infections as a means of survival [9]. Bacteria within biofilms are partially shielded from mechanical and shear stresses as well as environmental variables including altered pH, osmolarity, high pressure, excessive temperature, and nutrient starvation. In addition, the biofilm structure protects bacteria from external environmental conditions, antibiotics, disinfectants, and the host immune system [10]

Biofilm cells can exhibit physiological heterogeneity, including the persister cells and the viable but non-cultivable cells [11]. Microbial biofilms pose a therapeutic challenge, leading to persistent infections that are difficult to treat with traditional antibiotics [10].

Biofilm formation is a sequential phenomenon, involving a unique type of intercellular signaling called quorum sensing. It also requires the transcription of a distinct set of genes from those required for planktonic life [12]. The production of the extracellular matrix plays an important role in biofilm formation, protecting the cells from phagocytic cells and acting as a barrier to drugs and toxic substances. The viscoelastic properties of the EPS matrix are responsible for the mechanical stability of a biofilm. Several studies have identified the steps of biofilm formation. The old model, or 5-step model, is represented by all the steps described above separately [10]. In contrast, the new model or the inclusive model combines the micro-colony formation and the maturation, calling it growth, having only three steps [13]. Figure 1A illustrates the five-step process, whereas Figure 1B presents the inclusive model.

### 2.1. Initial Attachment to the Surface

The adhesion of a planktonic cell to a surface, biotic or abiotic, is the first step in the biofilm development process, which is dynamic and reversible, with the ability to reattach or detach from the surface during this phase [8]. Microbial cells may also use physical forces, such as van der Waals forces and electrostatic interactions, to adhere [14]. In the case of Gram-negative bacteria and some Gram-positive bacteria, fimbriae are a key feature that enables the adhesion of bacterial cells both to each other and to other surfaces [15,16]. The bacterial adherence to a surface is also significantly influenced by other factors like ionic strength, temperature, and pH [17]. In the process of biofilm formation, microbial cells adhere to surfaces and interact with each other within the community. This process is known as cohesion [14]. The connection between the bacteria and the surface is strengthened by the fimbriae, pilli, and flagella of the bacteria [18]. As the hydrophobicity of the surface reduces the repulsive forces between the bacteria and the surface, it could potentially contribute to the microorganism’s stronger adhesion. In contrast to hydrophilic and polar surfaces like metals and glass, microorganisms are more likely to adhere to hydrophobic and non-polar surfaces such as teflon and other plastics [14].

### 2.2. Formation of Micro-Colonies

Microbial cells begin to proliferate and divide after adhering to a biotic or abiotic surface [14]. This process is initiated by a specific chemical signaling within the EPS matrix [19]. Micro-colonies are then formed as a result of this process. The bacterial colonies of a biofilm typically contain a variety of micro-communities, which can collaborate with each other in multiple ways, through the exchange of substrates, the distribution of key metabolic products, and the excretion of metabolic end products [19].

### 2.3. Maturation of the Biofilm

During this phase, cell-to-cell communication is a crucial step in achieving the required microbial cell density. As a result, signaling molecules called autoinducers are secreted. These autoinducers facilitate quorum sensing [14]. Quorum sensing is a density-based mechanism that bacteria use to communicate chemically with each other [8]. During this stage, cells begin to produce an adhesive matrix that enables cells to stick to one another to form a multiplayer biofilm [14]. This matrix, or EPS, consists mainly of exopolysaccharides, protein, and DNA, and forms the three-dimensional structure of the biofilm, resulting in the formation of interstitial spaces in the matrix [20]. The water-filled channels act as a circulatory system, distributing essential nutrients and eliminating waste from the communities of micro-colonies within the biofilm [14].

### 2.4. Dispersion and Detachment

In order to move from a sessile to a motile form, the microbial cells within the biofilm multiply and disperse rapidly during this phase. There is then a natural pattern of detachment. Some bacteria, on the other hand, do not synthesize extracellular polysaccharide and instead disperse their cells into the environment [14]. Mechanical stress can also occasionally play a role in this process, with numerous triggers, including changes in nutrition availability, fluctuations in oxygen levels, an increase in hazardous compounds, or others [8]. Different saccharolytic enzymes produced by the microbial communities within the biofilm aid in the detachment process by releasing the microorganisms’ surface to a new location for colonization. To enable the bacteria to migrate to a new location, microbial cells at this stage upregulate the production of proteins involved in the development of flagella. Infections spread through the detachment and migration of microbial cells to other locations [14].

## 3. Current Understanding of Biofilm Formation in Animal Diseases

Bacterial biofilm-associated infections are a major challenge in the management of animal diseases in various sectors, including farm, husbandry, domestic, and wild animals. Understanding the types of biofilm-related infections prevalent in these animal populations is essential for the implementation of targeted preventive and therapeutic strategies. This topic explores the diverse spectrum of biofilm-related infections in animals, the main ones being dermatological, respiratory tract, urinary tract, and gastrointestinal infections, and their implications for veterinary medicine and animal husbandry practices.

Biofilm formation on the skin and mucosal surfaces of animals can lead to dermatological infections characterized by chronic inflammation, ulceration, and tissue damage. Common pathogens involved in these infections include *Staphylococcus* spp., and *Pseudomonas aeruginosa*. Dermatological biofilm-related infections are particularly prevalent in companion animals, contributing to skin disorders and wound complications [6,21].

In the respiratory tract, biofilm formation can lead to chronic respiratory infections, exacerbating respiratory diseases and impairing lung function. Bacteria such as *P. aeruginosa*, *Klebsiella pneumoniae*, and *Streptococcus* spp. are known to form biofilms in the respiratory mucosa of animals, leading to bronchitis, pneumonia, and lung abscesses. Respiratory biofilm-related infections are of significant concern in livestock farming and captive animal facilities [3,22,23].

It is also important to highlight that the accumulation of biofilms on bladder epithelium and renal surfaces predisposes animals to recurrent urinary tract infections. Pathogens such as *Escherichia coli*, *Proteus mirabilis*, and *Enterococcus* spp. are commonly implicated in urinary biofilm-related infections in animals. These infections pose challenges in animal husbandry and companion animals [24,25].

Biofilm formation within the gastrointestinal tract of animals can result in chronic enteric infections. Bacteria such as *E. coli*, *Salmonella* spp., and *Clostridium difficile* can form biofilms on intestinal epithelial surfaces, leading to gastrointestinal biofilm-related infections. These infections are prevalent in livestock farming, particularly in intensive farming systems and captive animal facilities, contributing to economic losses and public health issues [26,27].

Furthermore, biofilms facilitate bacterial adaptation to environmental pressures, accentuating the need for targeted interventions. With over 40% of human and livestock diseases attributed to biofilm-related infections, veterinary practitioners and animal husbandry professionals play a pivotal role in disease surveillance and management through effective biosecurity measures and antimicrobial stewardship [28,29].

By addressing these infections comprehensively, we can enhance animal health and welfare while mitigating the broader medical and economic impacts associated with biofilm-related diseases. Table 1 shows the infections associated with biofilm formation in companion, livestock, husbandry, farm, and wild animals.

### 3.1. Domestic Animals’ Biofilm-Related Infections

#### 3.1.1. Auditory System

##### Canine Otitis Externa (OE)

*Staphylococcus pseudintermedius* and *P. aeruginosa* are frequently the pathogens responsible for canine otitis externa (OE) in dogs [92]. This disease is associated with the development of biofilm [93]. Furthermore, cytological smears stained with periodic acid–Schiff (PAS) and modified Wright’s stain also support the presence of biofilm [3].

#### 3.1.2. Urogenital System

##### Urinary Tract Infections (UTIs)

The pathogenesis of urinary tract infections (UTIs) in dogs and cats is firstly driven by *E. coli*, and secondly associated with *Staphylococcus felis* [94]. These types of bacteria can live in a planktonic or biofilm state. They can grow on biologic or inert surfaces, such as the urothelium of the lower urinary tract or urinary catheters and surgical implants, respectively. Therefore, dogs and cats with ureteral stents and subcutaneous ureteral bypass systems are challenged by these biofilms. The uropathogenic *E. coli* (UPEC) are the most extensively studied bacterial biofilms in the urinary tract. Their pathogenesis starts in the bladder, through binding to uroepithelial cells via uroplakins and a3b1 integrins, activating the influx of neutrophils into the bladder lumen [25]. It is also important to focus on another species, *P. aeruginosa*, responsible for urinary tract infections in dogs, which can grow in a sessile community structure that confers protection against antibiotics, host defense mechanisms, desiccation, and ultraviolet light, as well as disinfectants [24,95,96].

##### Pyometra

Pyometra is a suppurative infection with the accumulation of a purulent exudate in the uterine lumen. In companion animals, pyometra is usually acute and life-threatening, requiring surgical treatment (ovariohysterectomy). The pathogenesis of pyometra remains unknown; however, hormonal conditions and bacteria virulence contribute to endometrial changes. *E. coli* is the most frequent bacterial species isolated from pyometra in companion animals and is associated with severe clinical cases. Other bacterial genera, even at lower frequencies, have also been involved in pyometra infections, such as *Staphylococcus* spp., *Streptococcus* spp., *Pseudomonas* spp., *Proteus* spp., *Enterobacter* spp., *Nocardia* spp., *Pasteurella* spp., and *Klebsiella* spp. A study demonstrated that *E. coli* strains isolated from samples of animals with pyometra infection produced components of the extracellular matrix, and analyzed the biofilm formation capacity of them. The results demonstrated that almost all *E. coli* isolates (93.3%) were able to produce biofilms [36]. Additionally, it was proven by Fiamengo et al. that most *E. coli* pathotypes present in canine pyometra are capable of biofilm production [37]. Conversely, Rocha et al. isolated 21 bacterial species from the uterine and vaginal contents of female dogs with pyometra and these bacterial isolates had low biofilm production [38].

#### 3.1.3. Integumentary System

##### Pyoderma (Skin Infection)

Pyoderma is a skin infection that affects pets, particularly dogs and cats, although the prevalence is lower in the latter group, ranging from 4% to 20% [97,98,99,100,101]. This condition is mainly caused by coagulase-positive staphylococci (CoPS). In dogs, *S. pseudintermedius* is the most prevalent microorganism, responsible for over 90% of the occurrences, while the other two CoPs species most associated with this skin infection are *S. aureus* and *S. coagulans* [97,102]. In cats, this illness is caused by *S. pseudintermedius*, *S. aureus*, or coagulase-negative staphylococci. Mariana Andrade and colleagues assessed the capacity of CoPs involved in skin infections in companion animals (*S. pseudintermedius*, *S. aureus*, *S. coagulans*) to produce biofilms, an important virulence factor that allows these bacteria to be more successful in promoting infections [21]. Therefore, it was demonstrated that there was a high production of biofilms by CoPs species from skin infections. Moreover, this biofilm production was mostly encountered in *S. pseudintermedius* and *S. aureus* clonal lineages associated with a high burden of antimicrobial resistance.

##### Wound Infections

Biofilm-infected wounds in dogs were first reported from a dog with chronic nonhealing pressure wounds. In this study, Swanson and co-workers identified in pressure wounds *S. intermedius*, *S. epidermidis*, and *S. canis* through 16S rRNA fragment sequencing and several bacterial types and no fungal species using pyrosequencing [42]. Other studies detected the presence of a mixed biofilm of canine tissue samples in 91 historical formalin-fixed and paraffin-embedded samples from dogs (n = 68), cats (n = 15), and horses (n = 8). From mixed biofilms, the authors suggested staphylococci or streptococci as the most predominant bacteria based on bacterial shapes, sites, and Gram-positive staining. However, further studies are needed to determine the bacterial species involved in the fullness [43]. Still, the same authors performed another study over mixed biofilms of dogs with postoperative surgical site infection. In this case, they identified the bacterial families *Porphyromonadaceae*, *Deinococcaceae*, *Methylococcaceae*, *Nocardiaceae*, *Alteromonadaceae*, and *Propionibacteriaceae* in the majority, resorting to a Next Generation Sequencing Analysis [43].

#### 3.1.4. Gastrointestinal System

##### Periodontitis

Bacterial periodontal disease is common in companion animals, causing severe oral cavity inflammation and a strong immune response. *Staphylococcus aureus*, *Streptococcus pyogenes*, and *Enterococcus faecalis* can colonize the tooth root canals, adhere to dentin walls, and frequently cause periodontitis in dogs. Dental biofilm, or plaque, plays a major role in the onset of dental caries, with the oral cavity’s moist environment and the adherent surfaces fostering plaque formation, which is difficult and expensive to remove. *Staphylococcus* spp. are frequently isolated from dog dental plaques. Most biofilm-forming bacteria originate in the bacterial plaque formed on the tooth surface. The identification of biological agents and efficient control of biofilm formation and consequently dental plaque are of constant concern in the veterinary practice [30].

#### 3.1.5. Respiratory System

##### Nosocomial Infections

*P. aeruginosa* is responsible for both local and systemic infections in dogs and cats. More predominantly, it causes skin, systemic, and urinary tract infections, and can also provoke respiratory infections [103]. Płókarz et al. analyzed 271 isolates of *P. aeruginosa* from dogs (external auditory canal, respiratory tract) and cats (nasal cavity, external auditory canal) with symptoms to determine the prevalence of five virulence genes (*pelA*, *pslA*, *ppyR*, *fliC*, and *nan1*) implicated in biofilm formation. The gene *ppyR* was the most frequent virulence factor identified, with 97.4% of prevalence [45]. Of all the strains tested, 90.6% and 86.4% from dogs and cats, respectively, were able to form biofilm. *Klebsiella* spp. is an important pathogen in animals and its prevalence has increased over time. In a recent study, *Klebsiella* spp. were isolated from ill dogs and cats, and 20% of the isolates were associated with respiratory infections [46].

### 3.2. Farm/Husbandry and Wild Animals’ Biofilm-Related Infections

#### 3.2.1. Reproductive System

##### Bovine Mastitis

Mastitis is a prevalent and costly illness on dairy farms, typically caused by *Staphylococcus* spp., with various species resulting in different clinical outcomes. *S. aureus* is considered to be an important etiological agent of bovine mastitis [54]. The ability of *S. aureus* to form biofilms, with the involvement of biofilm-associated proteins (Bap), provides an advantage in persisting within the bovine udder [13,75]. It is also important to highlight that in recent years, *S. aureus* biofilms have gained recognition as a major contributor to various infections, including chronic infections.

Moreover, the mammary gland is susceptible to colonization by other pathogens such as *Streptococcus* spp., *E. coli*, and coliform species, leading to parenchymal inflammation and disease. Mastitis can manifest in subclinical and clinical forms, impacting milk characteristics, quality, and sanitation, while also contaminating milking instruments, increasing the risk of zoonosis [62,63,64,65,75].

##### Endometritis in Dairy Cattle

Endometritis in dairy cattle is commonly caused by infection with *Trueperella pyogenes*, which can lead to hysteritis, endometritis, mastitis, liver abscesses, suppurative arthritis, and pneumonia in cattle. During calving, the opening of the cattle uterus may allow pathogenic bacteria such as *T. pyogenes* to enter the uterus through the birth canal. Studies have shown that *T. pyogenes* can invade host cells and become resistant to antibiotics by forming biofilms [78,79]. In some cases, *T. pyogenes* biofilm production may promote infection development, making it crucial to find a therapeutic option for reducing this bacterial property [79].

#### 3.2.2. Gastrointestinal System

##### Enteric Colibacillosis—Camel Calves

Camel calves are highly susceptible to bacterial infections, particularly those caused by *E. coli*. In fact, colibacillosis in young camels is the main cause of economic loss associated with poor growth, medication costs, and animal death. Neonatal diarrhea, caused by pathogenic *E. coli* expressing the *f17* gene, which encodes F17 fimbriae, has become a leading cause of morbidity and mortality in camel calves under three months of age, leading to significant losses in camel livestock [104]. In the biofilm formation process of *E. coli*, the key event is the attachment to the surface, leading to subsequent aggregation and mature biofilm formation. This increases the stability of bacteria to cause diseases and enhances their drug resistance capacity [105]

It has been demonstrated that *E. coli* isolates recovered from diseased animals revealed a high propensity to produce biofilm, suggesting the importance of biofilm-forming ability in the pathogenesis process [105,106]. In addition, several factors, such as different extracellular appendages, are involved in *E. coli* surface colonization. Their expression and activity are tightly regulated in space and time to ensure successful events, leading to mature biofilm formation [105,107].

##### Poultry Salmonellosis

Salmonellosis in poultry is a common occurrence in both domestic and wild birds. *Salmonella* can cause illness and death, particularly in very young chickens up to two weeks old. Symptoms can vary and include weakness, loss of appetite, poor growth, and watery diarrhea. In adult poultry, disease is rarely seen, even if they have bacteria in the blood. These animals can be infected with many different types of *Salmonella*; but the most important are *S.* Typhimurium and *S.* Enteritidis, which can induce clinical signs in poultry. Various *Salmonella* spp. are strong biofilm producers. Biofilm formation by *Salmonella* spp. has been shown to play a significant role in its pathogenicity due to its high resistance to antimicrobials and contributes to the increased virulence of the bacteria, thereby establishing a chronic infection [85,108]. But still, little is known about *Salmonella* biofilm assembly, making the prevention of the disease a challenge in the poultry production chain.

##### Clostridial Necrotic Enteritis

Clostridial necrotic enteritis (NE) is a serious gastrointestinal disease in poultry and avian species, caused mainly by *Clostridium perfringens* type A. This condition can lead to decreased growth performance, reduced feed efficiency, depression, anorexia, severe morbidity, and significant mortality in both young and adult birds. Recent research has shown that strong biofilm-producing isolates of *C. perfringens* have been identified from clinical sources, and these biofilms may play a role in the development of gastrointestinal diarrhea in animals. This is because they can promote bacterial survival and persistence in the small intestine during antibiotic treatment [26].

##### Clostridial Enterocolitis—Horses

Clostridial enterocolitis in horses can range from mild, self-limiting diarrhea to acute fulminant hemorrhagic diarrhea, which can be fatal for adult horses and foals. The disease is typically caused by *C. perfringens* type A. In the early stages of the disease, foals may present with anorexia, diarrhea, depression, and dehydration. Intestinal hypomotility or paralytic ileus may also be present. Although *C. perfringens* can be controlled with antibiotics, there is an increasing pressure of antibiotic resistance. In addition, bacterial biofilms act as a shield to protect the bacteria from antibiotics by decreasing their susceptibility to them [84]. Biofilm formation by *C. perfringens* is a major issue in veterinary medicine because it can lead to the adhesion of bacteria to surfaces in livestock farms and slaughterhouses. This can result in the contamination and colonization of new surfaces and the transmission of bacteria to other animals and humans [26].

#### 3.2.3. Nervous System

##### Swine Meningitis

Meningitis is a relatively common disease of young pigs in which infection leads to the inflammation of the sacs that surround the brain (meninges), causing disturbances in the nervous system. The disease causes high mortality and morbidity on pig farms and has an increasing zoonotic potential worldwide. It can be caused by a wide range of bacteria that can enter the bloodstream through wounds, tooth roots, the navel, and the tonsil, circulate in the body, and colonize the brain. *Streptococcus suis* is the most important bacterial agent that causes meningitis in pigs. Disease caused by *S. suis* is more prevalent in nursery pigs, but sucklers and young fatteners can also be affected [109,110]. The ability of several pathogenic microorganisms to form biofilms on host surfaces contributes to their virulence, and recent studies have found that *S. suis* can protect itself by forming biofilms, since the ability of bacteria to attach and colonize host tissues is a critical step in the initiation of infection, leading to increased drug resistance and prolonged disease [80,111].

##### Glässer’s Disease

Glässer’s disease is considered to be an important infection with worldwide distribution, causing considerable economic losses even on farms with a high health status. *Haemophilus parasuis* is the causative etiological agent of Glässer’s disease in pigs. This bacterium colonizes healthy pigs, and, under certain circumstances, some strains can invade the host and cause severe lesions. Systemic invasion is characterized by fibrinous polyserositis inflammation, polyarthritis, and fibrinous meningitis, and causes significant losses to producers due to reduction in weight gain, increases in the use of drugs, dead animals, and carcass depreciation [112]. Although the role of biofilm in *H. parasuis* pathogenesis is not clear, the expression of genes with putative function in biofilm formation has been detected, which plays a crucial role in the pathogenesis of the disease [22].

##### Hemorrhagic Septicemia—Bovines

Hemorrhagic septicemia is a severe and acute septicemic disease of buffalo and cattle, caused mainly by *Pasteurella multocida*. This disease leads to significant economic losses for livestock farmers due to its annual outbreaks with high mortality rates. Recently, a bioinformatic study revealed the presence of genes involved in strong biofilm-formation capacity in *P. multocida* strains. This suggests that these genes play a crucial role in allowing the bacterium to evade the host immune system and survive in hostile conditions as an adaptation [47].

##### Avian Colibacillosis

Avian colibacillosis, caused by avian pathogenic *E. coli* (APEC), is responsible for severe respiratory and systemic infections, which are a major cause of economic losses in the poultry industry worldwide. *E. coli* is the causative agent of several critical poultry diseases, including airsacculitis, pericarditis, peritonitis, salpingitis, polyserositis, colisepticemia, diarrhea, synovitis, osteomyelitis, and swollen head syndrome, collectively referred to as colibacillosis. This disease results in detrimental economic losses for the poultry sector due to morbidity, mortality, reduced body weight gain, carcass contamination, and recalled products [50]. Biofilm formation is an essential process in bacterial infection that leads to host disease, and APEC biofilms cause chronic, persistent, and recurring infections, making treatment difficult [48,49].

#### 3.2.4. Respiratory System

##### Porcine Respiratory Disease Complex

Swine respiratory diseases, often referred to as the porcine respiratory disease complex (PRDC), are prevalent in today’s pork production worldwide. PRDC is a multifactorial syndrome affecting the respiratory system of pigs, and environmental factors and management practices can trigger PRDC pathogens to cause severe health problems in postweaning and weaning-to-finishing pigs. PRDC is often associated with bacteria such as *Actinobacillus pleuropneumoniae*, *S. suis*, *P. multocida*, *Bordetella bronchiseptica*, *Gläesserella (Haemophilus) parasuis*, and *Mycoplasma hyopneumoniae*, which often operate in complex associations known as biofilms. These associations are responsible for maintaining the biogeochemical biosphere and, in some cases, can cause serious illness and its persistence in the host [22,23].

##### Hemorrhagic Pneumoniae—Minks and Foxes

Hemorrhagic pneumonia (HP) is a severe and often fatal illness that affects minks and foxes. It is caused by the bacterium *P. aeruginosa*, which is known for its ability to outcompete other organisms through various mechanisms. For instance, *P. aeruginosa* can form a polysaccharide-encased community known as a biofilm that can resist predation by protozoa [24].

##### Porcine Atrophic Rhinitis

Atrophic rhinitis is a widespread and economically important swine disease caused by *P. multocida* and *B. bronchiseptica*. The disease is characterized by the atrophy of the nasal turbinate bones, resulting in a shortened and deformed snout in severe cases. The *P. multocida* toxin and *B. bronchiseptica* dermonecrotic toxin are believed to interfere with the osteogenesis of the turbinate bone by inhibiting osteoblastic differentiation and/or stimulating bone resorption by osteoclasts [113]. A study has demonstrated the direct impact of biofilm formation on *B. bronchiseptica* pathogenesis in relation to porcine atrophic rhinitis and another one showed that biofilm formation by *P. multocida* may contribute to chronic infection and asymptomatic carriage [52].

#### 3.2.5. Cardiovascular System

##### Bovine Myocarditis

*Histophilus somni* is the most common cause of myocarditis in cattle, resulting in sudden death. Biofilm formation by *H. somni* is prominent in the cardiac tissue of myocarditis cases, with studies suggesting that the anaerobic environment in the myocardium is the reason for the pronounced biofilm formation in this tissue [3].

#### 3.2.6. Skeletal System

##### Osteomyelitis

Osteomyelitis in animals is primarily caused by infections that are either traumatic, surgical, or hematogenous in nature. For example, horses, pigs, broilers, turkeys, dogs, and cats are commonly infected with *Staphylococcus* spp., *Streptococcus* spp., *E. coli*, and other Gram-negative bacteria. In pigs, *Erysipelothrix rhusiopathiae* is also commonly found, while *T. pyogenes* is more prevalent in cattle. Osteomyelitis is a challenging condition to treat and typically requires extended antibiotic therapy and multiple surgical interventions to address the biofilm infection. In this condition, biofilms are characterized by colonies of loosely packed cocci embedded in an opaque matrix, as demonstrated by recent research [3].

## 4. Biofilm Tolerance/Resistance to Traditional Antimicrobials

The ability of microorganisms to form biofilms is closely linked to their tolerance and resistance to the traditional antimicrobials, as they can survive extremely high concentrations of antibiotics. This can be a problem, leading to long-lasting infections that are difficult to treat. Biofilms are a form of self-defense that allow a higher survival rate in hostile environments that are not optimized for the proliferation of these microorganisms [114]. By forming biofilms, cells can remain anchored as a polysaccharide matrix, and attached where nutrients are more available or regularly replenished, such as animal tissues [115]. In addition, biofilms enable bacteria to cohabit in close contact with each other, which increases their survival opportunities by promoting the exchange of nutrients and genetic elements, as well as enhancing cell-to-cell communication [116].

These features take biofilms to another level when compared to the nature of planktonic cells [117]. This makes them much less sensitive to harmful environmental agents and more resistant to chemical, physical, and biological factors [118]. In this way, biofilms can resist phagocytosis and only the cells adhered to their surface are removed [119]. Based on these properties, it is important to understand how bacteria use tolerance and resistance mechanisms to adapt to the unfavorable conditions imposed by the external environment [118]. In fact, diseases associated with the biofilm formation by pathogenic microorganisms are not easily solved by the host’s immune defenses, becoming persistent infections that respond poorly to antimicrobial treatments [120].

Microorganisms are naturally predisposed to tolerance, which is induced by environmental conditions, and the ability to survive being exposed to the toxic effects of a bactericidal agent [121]. Tolerance is generally a multifactorial phenomenon and is associated with several aspects, such as restricted growth at low oxygen levels, the presence of persistent cells, restricted access to antimicrobial substances, and the expression of biofilm-specific genes [122]. Normally, these microorganisms grow more slowly and have a longer stationary phase, which prevents the bactericidal agent from exerting a downstream harmful effect, even if the toxic substance is bound to the target [123]. Therefore, some of the mechanisms associated with tolerance include antibiotic-induced oxidative stress responses, a reduction in the growth rate, and the maintenance of persistent cells [118]. As the tolerance of biofilms to antimicrobial agents is related to their growth pattern, it is important to recognize that this tolerance increases as the biofilm matures [123]. Consequently, if the bacterial species that make up the biofilm were cultured planktonically, they would be expected to become susceptible to antimicrobial compounds again [124].

In contrast, resistance occurs when microorganisms are able to grow in the presence of a bactericidal or bacteriostatic compound at a concentration that would be inhibitory in other cases [123]. As the inherent tolerance of microorganisms in the biofilm promotes their survival in the presence of antimicrobial agents, this can lead to the development of resistance by increasing mutation rates, with lesions in mismatch repair or the emergence of persistent cells [125]. Resistance is a condition enhanced by mutations that make the bacterial cell impenetrable to the antibiotic [118]. As such, resistance can be associated with several mechanisms, including, for example, matrix β-lactamases or antibiotic efflux pumps [126]. Thus, resistance prevents an antimicrobial agent from interacting with the target and is typically specific to each antibiotic or its class [123]. Although resistance is usually driven by acquired mutations, many of which are associated with genes that can be carried out between bacteria, it can also be intrinsic, relying on the inherent properties of certain species or cells and the wild-type genes they carry [127].

In fact, the concepts of tolerance mechanisms, which involve survival in the presence of an antimicrobial agent, and resistance, involving growth due to the inhibition of the action of the bactericidal or bacteriostatic compound, are not independent from each other. Accordingly, there is some disagreement in the literature regarding certain mechanisms, with some authors considering one of them as a tolerance mechanism and others as a resistance mechanism [118]. To avoid this, some authors use the term ‘recalcitrance’ to describe the reduced susceptibility of biofilm cells to the action of antimicrobial compounds [128]. Therefore, regardless of the category attributed, it is necessary to investigate the mechanisms that make it difficult to treat bacterial infections associated with the development of biofilms, due to the low efficacy of traditional antimicrobial drugs in animals (Figure 2).

### 4.1. Mechanisms of Tolerance and Resistance

#### 4.1.1. Prevent Access to the Target

##### Extracellular Polymeric Substances

Biofilms are mainly composed of microbial cells, and extracellular polymeric substances (EPSs), which are composed of polysaccharides [129]. An EPS is also highly hydrated and can incorporate large amounts of water into its structure through hydrogen bonds, while also having the ability to acquire hydrophobic properties, depending on its organization [129]. The production of EPS varies depending on the microorganism, and can be enhanced by nutritional growth status, nitrogen, potassium or phosphate limitation, excess carbon availability, and its reduced growth rate [129]. These EPSs enhance the tolerance and resistance properties of biofilms by preventing the mass transport of antibiotics through the biofilm [130]. In *P. aeruginosa* biofilms, the structure consists of a large amount of EPSs, which act as a physical barrier that limits the penetration of antibiotics such as tobramycin into the deeper layers of the biofilm. This reduces the effectiveness of tobramycin in eradicating *P. aeruginosa* infections [131,132].

##### Failure of Antibiotics to Penetrate Biofilm

The suppression of antibiotic diffusion is mainly attributed to the matrix acting as a barrier; however, other factors are also involved. It should be noted that their limitation depends on different circumstances such as the biofilm growth conditions, the bacterial species involved, and the antimicrobial agent used [133,134]. The antibiotics that slowly penetrate in the biofilm lead to an adaptive phenotypic response that reduces the susceptibility of the microorganisms to the antibacterial agent before reaching lethal concentrations [135]. In addition, in cases where biofilm is associated with the medical devices, it grows on the retention sites of the medical devices [136], protecting the microorganisms from the action of antibiotics, which is currently associated with chronic veterinary diseases [137].

##### Extracellular DNA

Extracellular DNA (eDNA) is one of the major constituents of the EPS matrix and has multiple origins, ranging from quorum sensing controlled by bacterial secretion to cell death induced by phage activity or altruistic autolysis of subpopulations [138]. This biomolecule supports motility, provides structural stability to the biofilm, plays a broad role in cell adhesion, and acts as a protective mechanism against the host immune system and antimicrobial agents [139]. The eDNA is directly related to the reduced activity of antibiotics on the biofilm due to its anionic properties [140]. Capable of chelating cations, this molecule allows the formation of an ion-limited environment, such as the magnesium ion, acidifying the environment and triggering the action of signaling pathways that enhance resistance to antimicrobial agents [140].

##### Outer Membrane

The cytoplasmic membrane acts as a barrier between the extracellular environment and the cytoplasm of the organism, which is flexible due to its lipid composition. The permeability of this membrane is directly influenced by its fluidity, the reduction in which would have detrimental effects on the activity and structure of various membrane proteins present in the bilayer. In addition, to overcome this limitation, some bacteria develop additional external structures, such as a thick layer of characteristic peptidoglycan, which acts as a permeability barrier [141].

Due to the relative impermeability of the outer membrane, it is worth considering the presence of specific channels, such as porins, which can decrease the influx of drugs through various mechanisms, such as charge repulsion, size limitation, and hydrophobicity [142]. Therefore, it is important to infer that the outer membrane of bacteria slows down the permeation of some molecules into it, but does not completely prevent their influx and does not in itself lead to relevant levels of resistance.

In biofilms of *P. aeruginosa*, the outer membrane has low permeability, which serves as a barrier against antibiotics like polymyxins. This impermeable outer membrane limits the entry of polymyxins into the bacterial cell, reducing their effectiveness in disrupting the bacterial membrane [131,132].

##### Efflux Pumps

Efflux pumps are protein transporters found in cytoplasmic membranes that promote the defense of microorganisms against the action of antimicrobial agents by expelling toxins, including antibiotics, from the intracellular space [122]. When overexpressed, and because these transporters are easily altered by mutations acquired by microorganisms, they can confer high levels of resistance to clinically relevant antibiotics and become capable of surviving in extreme conditions [143]. In this way, several studies have reported the genes responsible for these transporters, giving the microorganism protection against the antimicrobial agents [144].

Although efflux pumps are also active in planktonic bacteria, their overproduction in biofilms has a high impact on the emergence of multidrug-resistant infections [145]. In several pathogenic bacteria common in animal diseases, such as *E. coli*, *S. aureus*, and *K. pneumoniae*, the overproduction of efflux pumps decreases the penetration of hydrophilic drugs, acting as a protective mechanism against several agents that pose a risk to the bacteria’s survival [146]. For these reasons, it is important to understand the molecular mechanism behind the overexpression of the efflux pumps in order to modulate them [147].

In several infections caused by *E. coli*, efflux pumps are overexpressed, since some proteins are responsible for pumping antibiotics, as in the case of tetracycline, out of the bacteria cell [131,132].

#### 4.1.2. Environmentally Adapted Responses

##### Direct Modification of Antibiotics

Besides preventing antibiotics from entering the cell, bacteria also have mechanisms that allow them to destroy or modify toxic molecules, by promoting the enzymatic modification of the antibiotic to a non-toxic form in the EPS. Exotoxins present in the matrix are responsible for this, and the most common classes are transferases, hydrolases, lyases, and redox enzymes [148]. A well-known example of this mechanism is the β-lactamases secreted by *K. pneumoniae* biofilms, which destroy ampicillin and prevent it from reaching the cells [149]. Another example is the case of *S. aureus* that can produce β-lactamase enzymes, which degrade β-lactam antibiotics like penicillin. This enzymatic degradation renders the antibiotic ineffective against the bacteria [131,132].

##### Oxidative Stress Responses

Oxidative stress represents the mechanism that bacteria and biofilms develop to defend themselves against the action of reactive oxygen species (ROS), such as superoxide anions or hydrogen peroxide [150]. It should be noted that, in addition to their specific mechanism of action, antimicrobial molecules also have a lethal effect by inducing the production of toxic levels of ROS that increase cellular respiration rates [151]. However, the pathways involved are highly dependent on the environmental conditions, the bacterial species involved, and the bactericidal agent [152].

Despite that, it is equally relevant to note that oxidative stress is induced in biofilms regardless of the presence or absence of antimicrobial agents in the environment [153], and can be generated by metabolic processes such as environmental stress factors (ROS cascade) or by the host immune system [154]. It can also be triggered by the exposure of cells to ionizing radiation, which promotes the intracellular formation of ROS [155]. ROS are known to cause lethal damage to cells, which interferes with DNA, and extend the range of action to proteins and lipids that have essential functions in microorganisms [128], having an impact on the lifespan of the species [124].

In biofilms, bacterial cells not only acquire the ability to counteract oxidative stress, but also use it as a strategy for adapting to adverse environmental conditions. In this way, microorganisms begin to dominate different environmental niches and become less vulnerable to the action of antimicrobial agents [156]. ROS also affect the characteristics of bacteria, altering their structure, morphology, and physiology, and become part of a dynamic signal in many cellular pathways that regulate biofilm formation [154]. In addition, oxidative molecules in the biofilm are also thought to promote the overexpression of specific proteins that are part of the efflux pumps, reducing the action of antimicrobial agents on microorganisms [157].

##### Persistent Cells

Persistent cells can survive a high concentration of an antimicrobial agent in a state of non-growth and non-division, which can reduce the susceptibility of a biofilm to the effects of antimicrobial agents [158]. To suppress the microorganisms that colonize a surface in clusters, persistent cells are phenotypically tolerant, surviving under conditions where most of the population dies quickly [159]. Thus, persistent cells temporarily lose their ability to proliferate in favor of survival and, when conditions become favorable again, these cells restart their division, ensuring the maintenance of the biofilm [107].

In a nutrient-limited biofilm, bacteria can exist in an extremely low metabolic state, and spontaneously reach a state of dormancy that confers increased resistance to the action of antimicrobial agents [120]. Although there is no consensus, this suggests two origins for this phenotypic change: prior to treatment with antimicrobial agents, when the exponentially growing population contains a pre-existing fraction of dormant, spore-like cells that do not grow, or after the culture has entered the stationary phase [160].

Persisters are altruistic cells that ensure the survival of a population in the presence of a lethal antimicrobial agent, but they only become prominent in a dense cell population. In the early exponential phase, when there are few neighboring each other, none would benefit from these mechanisms, so the highest level of persistent cells is reached when the population reaches the stationary phase [161]. It is also important to understand that the benefit that regular cells have, despite not having this resistance to the action of antimicrobial agents, is the ability to quickly restart growth. For this reason, the stationary population is composed of several cell types, not all of which enter a protective state [161]. As a result, persistent cells ensure that the biofilm resists the action of antimicrobial agents and are among those responsible for chronic infectious diseases [162]. In these clinical cases, which have a high impact on veterinary medicine, once the antibiotic treatment is stopped, the persistent cells present in the matrix start to grow again and repopulate the biofilm, causing recurrent infections.

##### Swarming

Swarming is a type of motility that allows highly differentiated bacterial cells to migrate and, in the case of biofilm cells, is highly resistant to the action of antimicrobial agents [163]. It is thought that this behavior may also be associated with a transient multidrug resistance phenotype, although the mechanism is not yet well understood.

##### Quorum Sensing

As mentioned above, quorum sensing is a type of cell-to-cell signaling that regulates the pattern of bacteria behavior according to a wide variety of cellular processes through the release of molecules into the extracellular environment, known as the autoinducers (AIs) [164]. Bacteria can detect and respond to an increase in population density by overexpressing a specific set of genes, to regulate cellular processes such as the expression of virulence factors, tolerance to certain molecules, toxin production, and motility [165]. The quorum sensing of Gram-negative bacteria includes the expression of the autoinducer-2 (AI-2) system, where the gene products of *luxS* are collectively referred to as AI-2 molecules [166].

The quorum sensing systems are important for the resistance mechanisms of the bacterial cells that make up the biofilm, promoting its formation, and also the overexpression of efflux pumps [167]. It is therefore important to note that a lack of quorum sensing is associated with the formation of biofilms with weaker and thinner structures, lower EPS production, and more susceptibility to the action of antibiotics [145].

In *P. aeruginosa* infections, the quorum sensing is used to regulate the expression of virulence factors and biofilm formation. High cell density and quorum sensing activation in *P. aeruginosa* biofilms lead to the upregulation of genes encoding efflux pumps, which expel antibiotics like ciprofloxacin from the bacterial cells, contributing to antibiotic resistance [131,132].

##### SOS Response

The SOS response includes all the molecular mechanisms that work to respond to damage caused by chromosomal DNA, whether caused by radiation, oxidizing radicals, or antimicrobial agents [128]. This mechanism is linked to resistance mechanisms that guarantee the survival of bacterial cells [168]. It is known that the SOS response in a heterogeneous and nutrient-limited environment, such as the biofilm structure, confers high specific tolerance to antibiotics such as ofloxacin, due to the ability to respond to topoisomerase inhibition [168].

Upon exposure to antibiotics like methicillin, *S. aureus* can activate the SOS response, which targets cell wall synthesis. The SOS response facilitates DNA repair mechanisms, allowing *S. aureus* to overcome DNA damage caused by methicillin and survive antibiotic treatment, contributing to the development of methicillin-resistant strains [131,132].

#### 4.1.3. Physiological and Metabolic Heterogeneity with Nutritional Limitations

Physiological and metabolic heterogeneity emerges due to the nutrient and oxygen gradient inherent in the structure of a biofilm, leading to major changes in bacterial growth patterns [169]. This gradient is enhanced by the fact that cells close to the surface have access to more nutrient resources, preventing them from penetrating deeper into the biofilm. The same happens with the rate of oxygen, so that cells in deeper layers are deprived of oxygen [170]. It is therefore clear that most mature biofilms grow more slowly, due to the reduced access to oxygen and nutrients and the increased difficulty in waste removal [171]. Thus, since the growth rate and metabolic activity are affected by the availability of nutrients and oxygen in biofilms, it is possible to understand why the more peripheral regions are characterized by a wider proliferation of bacteria [172]. Since bacteria have reduced metabolic activity and growth rates in hypoxic conditions, antimicrobial agents are less effective locally in these regions [173]. Even when antibiotics reach the entire depth of the biofilm, the areas most susceptible to their action have been shown to be those with sufficient oxygen and high protein synthesis [174]. This effect is further supported by the fact that biofilms treated with antimicrobial agents under anaerobic conditions are more tolerant to their action than those under aerobic conditions [175]. It is also thought that hypoxia may promote the bacterial survival by reducing the production of ROS, which use the presence of molecular oxygen to induce the SOS response [176].

Furthermore, the stringent response is a highly conserved signaling pathway triggered during biofilm formation that promotes tolerance and resistance, as well as the emergence of persistent cells, in order to ensure survival under conditions of nutrient starvation [177]. This process leads to an increased production of the guanosine pentaphosphate and guanosine tetraphosphate, (p)ppGpp, which acts in response to external stress caused by the limitation of amino acids and carbon/fatty acid supplies [178]. Its accumulation is associated with a reduction in cellular protein synthesis activity, regulating the biosynthetic capacity of cells [179]. It was then suggested that the stringent response also contributes to tolerance in biofilms by reducing the effects of oxidative stress, as this signaling pathway prevents ROS-induced damage by positively regulating enzymes with antioxidant properties [180].

#### 4.1.4. Genetic Mechanisms

Genetic determinants are involved in biofilm formation, and there are genes responsible for reducing the susceptibility of the cells to the action of antimicrobial agents. Some of the widely described mechanisms of these characteristics are, for example, the differential permeability of the outer membrane and the presence of efflux systems, as described above. In addition to intrinsic mechanisms, bacteria can acquire tolerance or resistance to the action of specific antibiotics by inactivating the antibiotic or by modifying its target through post-translational changes or genetic mutations [143].

Horizontal gene transfer is one of the main factors involved in the acquisition of new traits that promote increased tolerance and resistance in the biofilms, based on the uptake of eDNA from the environment and the transfer of plasmids by conjugation. It has been suggested that the transfer of these plasmids is more efficient in biofilms, due to the spatial proximity of the cells and their sessile status [181]. It is also important to note that the number of copies of plasmids in biofilms is also associated with an increased rate of transmission of antibiotic resistance genes [182]. Integrons, genetic elements that promote the incorporation of gene cassettes into the bacterial genome, are also of particular importance [183]. These promote the spread of beneficial traits and their expression is increased in biofilms due to the stringent response, enhancing the prevalence of genes associated with decreased susceptibility to antimicrobial agents by horizontal gene exchange [184].

Different genes or regulators are involved in processes that increase tolerance, such as those that regulate the transition of cells to a persistent state [128]. However, while many are acquired through horizontal transfer, others arise through advantageous mutations that become widespread in populations. Combined with the fact that biofilm cells accumulate mutations at a higher rate, their mode of development promotes the emergence of permanently hypermutable strains [185]. In addition, biofilm cells are also more likely to spontaneously mutate because they are exposed to high levels of endogenous oxidative stress, which induces DNA damage [185]. As a result, multidrug-resistant strains are becoming increasingly common in clinical and veterinary settings, challenging conventional therapies and highlighting the need to develop and apply new treatments to suppress the proliferation of biofilms and limit their impact on chronic infections.

#### 4.1.5. Trained Immunity

Trained immunity, also referred to as innate immune memory, is acquired upon initial exposure to a stimulus prompting an immune response [186]. Within the innate immune system, there are memory-like responses to previous encounters with both microbial and non-microbial challenges [187]. Recent research has outlined two distinct hypotheses regarding the induction of adaptive immunity and tolerance in innate immune cells. The first, termed the stressor-dependent hypothesis, proposes that specific pathogen-associated molecular patterns (PAMPs) or damage-associated molecular patterns (DAMPs), such as β-glucan, BCG, oxidized low-density lipoprotein (oxLDL), and heme, trigger the induction of adaptive immunity. Conversely, a Gram-negative endotoxin (lipopolysaccharide or LPS) predominantly promotes tolerant reactions [188,189]. Conversely, the Gram-negative endotoxin (lipopolysaccharide or LPS) predominantly promotes tolerogenic responses [190,191]. The second hypothesis, the dose-dependent hypothesis, proposes a biphasic dose–response relationship: low-dose priming induces an adaptive phenotype, whereas high-dose exposure results in an immunosuppressive phenotype (tolerance) upon subsequent insult [188,189].

A recent study investigated the effect of LTA priming on murine bone marrow neutrophils in vitro and demonstrated its role in inducing distinct memory-like inflammatory responses, trained sensitivity, and tolerance in a dose-dependent manner [190]. The results showed that low-dose LTA-primed neutrophils exhibited elevated levels of pro-inflammatory mediators, indicative of trained sensitivity. Conversely, high-dose LTA-primed neutrophils induced an immunosuppressive phenotype characterized by decreased pro-inflammatory responses and increased IL-10 production [190]. Furthermore, another study demonstrated that resident dermal macrophages undergo local programming independent of bone-marrow-derived monocytes during staphylococcal skin infection, resulting in transient increased resistance to subsequent infections [191].

Conversely, gut-microbiota-derived small extracellular vesicles (EVs) may serve as a critical link between the immunomodulatory properties of the gut and neutrophils. A low concentration of EVs was found to induce the increased production of pro-inflammatory mediators. In contrast, neutrophils primed with high concentrations of small EVs displayed an immunosuppressive phenotype [192].

Recent investigations move long-term adaptive responses of the innate immune system into focus. As such, the potential of LPS to prevent animal-associated infections has been shown. An example is the udder infections with *E. coli*, which are a serious problem for the dairy industry. Günther and co-workers showed that a mild transient stimulation of healthy udders with a single low dose of LPS (1 μg/quarter) will not only reduce the severity of a subsequently elicited *E. coli* mastitis but will protect the udder from colonization with *E. coli* pathogens for three to ten days [193]. More recently, Lajqi and colleagues suggested the profound influence of preceding contacts with pathogens on the immune response of microglia [194]. The impact of these interactions—trained immunity or immune tolerance—appears to be shaped by the pathogen dose.

## 5. Combatting Antimicrobial Resistance: Alternative Therapies to Control Biofilm Formation—Case Studies

In response to the escalating levels of multi-resistance exhibited by traditional antimicrobial agents, innovative therapeutic approaches have been developed to effectively address and manage biofilm formation. These alternative strategies have proven crucial in veterinary settings, where the impact of antimicrobial resistance is significant. Additionally, the importance of in vitro studies cannot be overstated, as they serve as essential tools for elucidating the efficacy and mechanisms of these alternative therapies. This section provides a comprehensive overview of these novel therapies, and their applications in animals, and highlights the pivotal role of in vitro or in vivo studies in advancing our understanding of their potential impact and effectiveness. Table 2 summarizes emerging approaches to treat biofilm-associated infections.

### 5.1. Aptamers

Aptamers are a molecular alternative that has shown promising results in controlling microbial infections. Developed by a method known as the systematic evolution of ligands by exponential enrichment (SELEX), aptamers are small single-stranded oligonucleotides (typically DNA or RNA) that bind specifically to targets with high affinity and selectivity [195]. SELEX is an iterative in vitro methodology of several cycles of the incubation, partition, and amplification of an initial random oligonucleotide library that is consecutively restricted until a unique set with high affinity for the target molecule is obtained, being subsequently sequenced for the identification and characterization of potential aptamers [196].

Developed with the aim of being an alternative to antibodies, aptamers have several advantages such as lower complexity and immunogenicity and can be selected for virtually any type of molecule, including proteins, cells, small molecules, and toxic compounds [197]. They have greater thermal stability and can return to their functional conformation after denaturation without loss of activity [198]. Furthermore, simple chemical synthesis reduces production costs while allowing the introduction of modifications to improve/overcome some functional limitations (e.g., increasing resistance to enzymatic degradation by adding modified nucleotides, also known as nucleic acid mimics (NAMs)), or to adapt to specific applications by attaching active compounds (e.g., drugs) [199,200].

Although most aptamers are developed for diagnostic analytical applications, aptamers have already demonstrated therapeutic potential [200]. Antimicrobial activities of aptamers have been reported in several studies, including for biofilm control [200]. Using cell–SELEX methodologies, it is possible to select aptamers with affinity for biofilm-forming microorganisms and thus direct their functionality towards the complex biological structure that biofilms form [201,202]. Table 2 summarizes aptamers that have antibiofilm activity and/or potentiate the antimicrobial action of other agents such as antibiotics against pathogenic microorganisms.

**Table 2 pathogens-13-00320-t002:** Biofilm-associated infections in animals and major etiological agents.

Emerging Approaches to Treat Biofilm-Associated Infections	Biofilm- Associated Species	Source	Disorder/Infection	Mechanism	In Vitro Main Result	In Vivo Main Result	Reference
**Prebiotics**							
1% xylitol (XYL) with 1% galacto-oligosaccharides (GOSs) or 1% fructo-oligosaccharides (FOSs) or 1% isomalto-oligosaccharides (IMOs) or 1% arabinogalactan (LAG)	*Staphylococcus aureus*, *Staphylococcus epidermidis*	Private collection of Department of Pharmacy, University of Salerno, Italy	*S. aureus* infection in skin	GOS, FOS, IMO, and LAG in combination with XYL at 1% concentration modulate the skin microbiota and play a role in stabilizing indigenous beneficial strains and inhibiting pathogenic microorganisms. These combinations show selective species-specific activity in the planktonic and sessile phases of the strains studied and are able to enhance the prebiotic activity of XYL.	A predominant bacteriostatic effect in the planktonic phase and an overall reduction in *S. aureus* biofilm formation was observed for all tested formulations. The combinations of 1% XYL with 1% GOS or 1% FOS or 1% IMO or 1% LAG can help to control the balance of skin microbiota and are good candidates for topical formulations.	Not tested	[203]
Agave fructans (AF)	*Staphylococcus aureus*	Bovine subclinical mastitis diagnosed by California Mastitis Test	Bovine mastitis	AF might bind to *S. aureus* surface proteins and limit their growth.	The AF showed a decrease in maximum growth rate (µmax) and optical density max levels in all isolates with all concentrations. Zones of inhibition were observed due to the effect of all AF concentrations in all isolates in a dose-dependent manner. *S. aureus* biofilm formation was inhibited by all AF concentrations assessed in this study.	Not tested	[75]
**Probiotics**							
*Lactobacillus kefiri* 8321 *L. kefiri* 83113 *L. plantarum* 83114	*Salmonella enteritidis*, *Salmonella Typhimurium*, *Salmonella Gallinarum*	Chickens, poultry compost, and eggs	Paratyphoid *Salmonella*	Exclusion mechanisms and production of antibacterial compounds that interact with the components of the biofilm matrix or the pathogen.	Pre-incubation of SE 115 with the three *Lactobacillus* strains resulted in greater inhibition of biofilm formation compared to co-incubation. The surface proteins extracted from all three *Lactobacillus* strains were tested, indicating a significant inhibition of biofilm formation.	Not tested	[204]
*Lactobacillus* *acidophilus* LA5 *Lacticaseibacillus casei* 431	*Staphylococcus aureus*	Collection strain (ATCC 25923)	*S. aureus* infection	The exopolysaccharides and biosurfactants released in the CFS are associated with the high activity of biofilm removal.	On polystyrene, the cell-free supernatant of *L. acidophilus* removed 70.60% of the *S. aureus* biofilm, while *L. casei* removed 65.30%. On the glass, the CFS of *L. acidophilus* removed 87% of the biofilm. Overall, the cell-free supernatant from *L. acidophilus* exhibited greater biofilm elimination compared to *L. casei*.	Not tested	[205]
*Bacillus subtilis* PS216	*Camplylobacter jejuni* subsp. *Jejuni* strain	Collection strain (NCTC11168)	Foodborne infection (chicken meat)	The production of diffusible antimicrobial molecules and genes, such as bacillene, belonging to *B. subtilis*, leads to the inhibition of *C. jejuni* growth and disintegration of its biofilm.	The study demonstrated that under microaerobic conditions, *B. subtilis* PS-216 can disintegrate the pre-established biofilm of *C. jejuni* after 12 h of co-incubation. *B. subtilis* was able to disrupt the pre-established biofilm and significantly reduce *C. jejuni* colony counts.	Not tested	[206]
*Bacillus* spp.	*Staphylococcus aureus*	Cows with mastitis (belong to the “Mastitis Pathogens Culture Collection of Embrapa Dairy Cattle”) and ewe with mild mastitis	Bovine mastitis	The *nuc* and *aur* genes encode enzymes associated with biofilm dispersal. The mechanism of action may be linked to chemical modifications on the abiotic surface through bacterial polysaccharides.	*Bacillus* sp. 18, *B. altitudinis* 27, and *B. velezensis* 87 significantly reduced biofilm formation in most *S. aureus* strains without affecting bacterial growth by approximately 40%. *S. aureus* O46, a strain known for its strong biofilm production, was tested against the extracellular polymeric substances (EPSs) produced by *B. velezensis* TR47II. The results showed an 83% inhibition of the biofilm formation.	Not tested	[207]
*Limosilactobacillus fermentum*, *Lactiplantibacillus plantarum*	Enterotoxigenic *Escherichia coli*	“Laboratório de Microbiologia do Hospital Veterinário da Universidade Estadual de Santa Cruz”	Neonatal diarrhea	The lyophilized cell-free supernatant (CFS) method inhibits biofilm formation by producing bacterin surfactants that interact with and degrade the polymeric matrix, exposing the microorganisms.	CFS generated from *L. plantarum* (dose: 40 mg/mL) was effective in controlling biofilms, significantly reducing their production. It has also been observed that the direct use of *Lactobacillus* has a bioprotective effect through coaggregation with *E. coli* in vitro.	Not tested	[208]
*Limosilactobacillus reuteri* S5	*Salmonella* Enteritidis	Collection strain (ATCC 13076)	Salmonellosis	*L. reuteri* S5 significantly decreased the expression of adhesion and invasion genes and membrane and cell wall integrity genes of *S. enteritidis* ATCC 13076.	The results show that *L. reuteri* S5 has a high capacity to inhibit *S. enteritidis* biofilm formation compared to the control. Gene expression analysis showed that *L. reuteri* S5 can disrupt membrane and cell wall integrity and reduce the expression of *S. enteritidis* virulence factors.	Not tested	[209]
*Bacillus subtilis* KATMIRA1933 *Bacillus amyloliquefaciens* B-1895	*Salmonella enterica* subsp. *enterica* serovar Hadar, *Salmonella enterica* subsp. *Enterica serovar* Enteritidis phage type 4 and *Salmonella enterica* subsp. *enterica serovar* Thompson	NA	Salmonelosis (foodborne infection)	Compounds produced by *Bacillus*, such as organic acids, enzymes, and/or inhibitory substances similar to bacteriocins.	The biofilm formation of *Salmonella* Hadar, Enteritidis phage type 4, and Thompson strains was more affected when incubated with the *B. subtilis KATMIRA* 1933 by 51.1%, 48.3%, and 56.9%, respectively, than with the *B. amyloliquefaciens* B-1895, which showed values of 30.4%, 28.6%, and 35.5%, respectively.	Not tested	[210]
*Lactobacillus* *plantarum 22F*, *25F* *Pediococcus acidilactici 72N*	*Escherichia coli*	Feces or wastewater at a swine farm	*E.coli* infection	Biofilm dispersal can be contributed to by CFS components, such as enzymes or dispersal signaling molecules.	The non-neutralizing cell-free supernatants of *P. acidilactici* 72 N showed the greatest reduction in biofilm formation, with the percentages of inhibition ranging from 50.20% to 82.28%. Among the *E. coli* strains tested, *L. plantarum* neutralizing CFS 25F showed the highest potential for inhibiting biofilm formation. However, the maximum percentage of inhibition was obtained by *P. acidilactici* 72N (52.29%).	Not tested	[211]
**Postbiotics**							
*Lentilactobacillus kefiri LK1* *Enterococcus faecium EFM2*	*Staphylococcus aureus*, *Enterococcus* *faecalis*, *Pseudomonas aeruginosa*, *Escherichia coli*	Bovine mastitis milk	Bovine mastitis	*Lentilactobacillus kefiri* LK1 and *Enterococcus faecium* EFM2 downregulate key genes involved in biofilm formation.	The postbiotics *Lentilactobacillus kefiri* LK1 and *Enterococcus faecium* EFM2 were found to significantly inhibit the growth and biofilm formation of *S. aureus*, *P. aeruginosa*, *E. faecalis*, and *E. coli*. The hydrophobicity, self-aggregation, and EPS production of the pathogens were also significantly reduced by the postbiotics used.	Not tested	[212]
*Lactobacillus sakei EIR/CM-1*	Methicillin-resistant *Staphlococcus aureus* (MRSA), *Streptococcus agalactiae*, and *Streptococcus dygalactiae* subsp. *dysgalactiae*	Collection strain (ATCC 43300, ATCC 27956, ATCC 27957)	Ruminant mastitis	The HPLC analysis of *L. sakei* confirmed the presence of organic acid, which are secondary metabolites that inhibit the growth of pathogens and their biofilm production. *L. sakei* also secretes oleic acid, which may be responsible for its antibacterial activity.	Co-incubation was found to inhibit the biofilm formed by more than 85% in all pathogens. Pre-treatment reduced biofilm formation of *S. agalactiae* ATCC 27956 by 86,44% and of *S. dysgalactiae subsp. dysgalactiae* ATCC 27957 by 95.10%. Post-treatment was unable to abolish biofilm formation.	Not tested	[213]
**Aptamers**							
JN27 JN08	*Pseudomonas aeruginosa*	Collection strain (ATCC 14502)	NA	The aptamers were selected against the whole cell of *P. aeruginosa*, confirmed by SYTO9/PI (live/dead) staining of planktonically and biofilm grown cultures.	The aptamers bind to *P. aerugionosa*-biofilm-grown cells, but they do not have intrinsic bacteriostatic or bactericidal activity following a 15 min incubation with 1 μM aptamer.	Not tested	[214]
A16 A46 A1		Human clinical isolate	NA	The aptamers were selected against C4-HSL, an essential inducer of quorum sensing in the formation and survival of *P. aeruginosa*. The aptamers showed a high affinity and specificity for this molecule, being able to block its effect and thus prevent QS in biofilm-forming *P. aeruginosa*.	Biofilm inhibition experiments in vitro showed that the biofilm formation of *P. aeruginosa* was efficiently reduced by about 1/3 by the aptamers compared to the groups without the aptamers.	Not tested	[215]
NC2 NC5 NC1 NC6		Collection strain (ATCC 10145)	NA	In this study, DNA aptamers previously selected were used as a delivery system to deliver silver nanoparticles to the EPS matrix of *P. aeruginosa* biofilms.	Among the NC2, NC1, and NC5 aptamers, there was a decrease in *P. aeruginosa* biofilm formation (13.02%, 11.39%, and 11.21%, respectively) compared to the positive control and random aptamers used.	Not tested	[216]
PA-ap1		Collection strain (ATCC 27853)	NA	An aptamer named PA-ap1, which was selected for its ability to target *P. aeruginosa* cells, was used as targeting delivery system to enhance the efficiency of antibiofilm agents, including single-walled carbon nanotubes (SWNTs) and ciprofloxacin–SWNTs.	In vitro tests demonstrated that the aptamer–SWNTs could inhibit 36% more biofilm formation than SWNTs alone. Similarly, the aptamer–ciprofloxacin–SWNTs had a higher antibiofilm efficiency than either component or simple mixtures of two components.	Not tested	[217]
ALSap-5 ALSap-8		Reference strain (PAO1)	Human, infected wound	The aptamers were selected to bind and inhibit the function of N-acyl homoserine lactone (HSL), a signaling molecule of the quorum sensing system, with the aim of interfering with signaling and attenuating the virulence of *P. aeruginosa*, including biofilm formation.	In the presence of 0.05 µM ALSap-5, the formation of biofilm decreased by 20%. As the ALSap-5 concentration increased to 0.5 µM, about 90% of biofilm was inhibited. With ALSap-8 (6 µM), biofilm formation was also decreased by 9.3%. In addition, they did not affect bacteria growth, which meant that they may have slimmer risk of inducing bacterial drug resistance vs. traditional antibiotics.	Not tested	[218]
Aptamer 3	*Salmonella enterica* subsp. *enterica serovar Ccholeraesuis*	Human clinical isolate	NA	Aptamer was selected against the whole cell of *S. choleraesuis*. Mass spectrometry analysis of the protein phase of the bacterial suspension after binding with the aptamer identified flagellin as the target molecule. A specific binding experiment with flagellin protein proved the high-affinity binding of the aptamer.	The mean thickness of the biofilm in the presence of aptamer 3 was significantly less than in the absence. In addition, survival of *S. choleraesuis* in the biofilm declined to 3.8% with the addition of 1.1 μM aptamer 3 and 5 μg/mL ampicillin sodium (synergic effect).	Not tested	[219]
		Collection strain (ATCC 10708)	NA		The optical densities of biofilm formation decreased to 28.36 ± 0.57% relative to the blank controls as well as survival ratio of established biofilm decreasing to 39.25 ± 1.18% relative to the controls with the addition of Apt3–ampicillin conjugate (synergic effect).	Not tested	[220]
ST-3	*Salmonella enterica* subsp. *enterica* serovar Typhimurium	Collection strain (CMCC 50115)	NA	A bifunctional conjugate was assembled by linking the ST-3 aptamer to graphene oxide (GO), combining the antibiofilm effect of the GO and the bacteriostatic effect of the ST-3 aptamer. In addition, ST-3 facilitated the entry of GO into the biofilm and decreased the potential of the cell membrane to prevent its growth.	*S. typhimurium* biofilms were inhibited 93.5 ± 3.4%, and 84.6 ± 5.1% of biofilms were dispersed by a ST-3-GO conjugate.	Not tested	[221]
SA31	*Staphylococcus aureus*	Collection strain (DSM 20231)	NA	Aptamers pre-selected against the whole cell of *S. aureus* were used as biofilm-targeting agents in a liposomal drug delivery system, allowing the liposomes to accumulate around the *S. aureus* biofilms and the subsequent release of a combination of antibiotics.	Aptamer-targeted liposomes encapsulating a combination of vancomycin and rifampicin were able to eradicate *S. aureus* biofilm upon 24 h of treatment.	Not tested	[222]
Aptamer 1		Human clinical isolate (MRSA strain)	NA	Graphene oxide (GO)-loaded aptamer/berberine bifunctional complexes were developed. Aptamer 1 was selected against penicillin-binding protein 2a (PBP2a) to reduce cell-surface attachment by blocking the function of PBP2a and berberine was used to attenuate the level of the accessory gene regulator (agr) system, which plays an important role in mediating MRSA biofilm formation.	Application of 200 nM aptamer 1 or 50 lg/mL berberine alone was able to inhibit MRSA biofilm formation by 20.8% and 41.2%, respectively. The inhibition rate declined to 70.3% when the berberine/aptamer 1 (containing 200 nM aptamer 1) complex was added. These results indicated that aptamer 1 can improve the targeting rate of berberine and has a synergistic inhibitory effect with berberine. Moreover, the inhibition rate peaked at 92.8% after treatment with the GO–berberine/aptamer 1 (containing 200 nM aptamer 1) complex, suggesting that GO can improve the stability and availability of the berberine/aptamer 1 complex in biological environments.	Not tested	[223]
S15K3 S15K4 S15K6 S15K13 S15K15 S15K20	*Staphylococcus aureus* *Escherichia coli*	Cow’s milk (strains BPA-12 and EPEC 4)	Subclinical mastitis	The six polyclonal DNA aptamers were selected simultaneously against *S. aureus* BPA-12 and *S. agalactiae* and *E. coli* EPEC 4, so they have binding affinity to both strains. It is hypothesized that they bind to the bacteria’s flagella and thus prevent initial attachment and subsequent biofilm formation.	Aptamer S15K6 showed the highest percentage of antibiofilm activity against *S. aureus* BPA-12 (37.4%), while aptamer S15K3, S15K4, S15K13, and S15K20 also showed strong inhibition percentage on *S. aureus* BPA-12. Aptamer S15K20 showed the highest percentage of antibiofilm activity against *E. coli* EPEC 4 (15.4%). Aptamers S15K13 and S15K20 showed antibiofilm activities against both *S. aureus* BPA-12 and *E. coli* EPEC4, and thus potentially have broad reactivity.	Not tested	[224]
SELEX 10 colony 5	*Escherichia coli*	Human clinical isolate (strain EPEC K1.1)	Diarrhea	Aptamer was selected against the whole cell of *E. coli.* The motility examination combined with qPCR was applied to prove that the aptamer was able to inhibit biofilm formation by interfering with the motility ability, which might be linked to the flagella function, and also by reducing the mRNA level of biofilm-formation-related genes, where the mRNA level of *motB*, *csgA*, and *lsrA* genes reduced significantly compared to the untreated group.	Aptamer SELEX 10 colony 5 exhibited the highest biofilm inhibition towards EPEC K1.1 shown by the lowest OD value. The aptamer was effective in killing EPEC K1.1 in a dose-dependent manner where this antibacterial activity was comparable to that of positive control (ampicillin) as measured by a cleared zone formation at an aptamer concentration of 1 μM.	Not tested	[225]
R8-su12	*Streptococcus suis* serotype 2	Ante-mortem blood culture from a pig (strain P1/7)	Meningitis	Aptamer was selected against the whole cell of *S. suis* but proved to bind also against other *S. suis* serotypes, i.e., 1, 1/2, 9, and 14. It is hypothesized that the aptamer targets the surface molecules on *S. suis* cells, affecting the biofilm formation.	The biofilm formation of *S. suis* SS 2, P1/7, cultured with R8-su12 RNA aptamer, was significantly reduced by 61.2% when compared to control (*p* < 0.05).	Not tested	[226]
AptBH	*Streptococcus mutans*	Collection strain (PTCC 1683)	NA	The aptamer selected to bind specifically to the *S. mutans* wall was coupled to silver nanoparticles, which, when they bind to the cell wall, cause the membrane to rupture, and the accumulation of peroxides causes the cell wall to oxidize and ultimately destroy the bacterium.	Silver nanoparticle–aptamer complex could inhibit biofilm formation in a dose-dependent manner. At a concentration of 100 mg/mL after 48 h, it inhibited 43% of the biofilm formation and degraded 63% of the formed biofilm.	Not tested	[227]
PmA2G02	*Proteus* *mirabilis*	Collection strain (MTCC 1429)	NA	In silico analysis revealed a higher probability of aptamer binding to *P. mirabilis* surface proteins. In addition, a significant reduction in swarming motility was observed when *P. mirabilis* was exposed to the aptamer, revealing a possible interaction with the proteins involved in this process (e.g., flagellin).	The application of the aptamer (3 μM) resulted in a significant reduction (23.3%) in biofilm formation compared to the control biofilms. Mature biofilms disintegrated, and the number of free-floating cells increased with addition of the aptamer (maximum effect with 3 μM).	Not tested	[228]
APG-1	*Porphyromonas gingivalis*	Human clinical isolate (strain IR-TUMS/BPG5)	NA	The aptamer was selected for the specific identification of *P. gingivalis* and was subsequently linked to nanographene oxide (NGO), forming a targeted NGO-carrying drug delivery system for antimicrobial photodynamic therapy (aPDT). Aptamer serves as a nucleic acid drug and a targeted delivery system for NGO.	Therapy using 1/2 × and 1/4 × MBC of DNA–aptamer–NGO plus irradiation of the diode laser light (1 min) has a significant antibiofilm effect against *P. gingivalis* in comparison with the control group (*p* < 0.05). Although diode laser and 1/4 × MBC of DNA–aptamer–NGO alone were not able to inhibit the biofilm considerably (*p* > 0.05), a significant biofilm degradation was observed in *P. gingivalis* biofilm treated with 1/4 × MBC of DNA–aptamer–NGO alone compared to the control group (*p* < 0.05).	Not tested	[229]
**Bacteriophages**							
LysKΔamidase	*Staphylococccus aureus*	Cows with bovine mastitis	Bovine mastitis	An engineered lysin was generated by fusing the N-terminal 220 amino acids with the C-terminal 105 amino acids of the staphylococcal phage lysin LysK. LysKΔamidase resulted from the remotion of the middle amidase catalytic domain. This lysin has a lytic activity.	LysKΔamidase disrupted biofilms produced by MRSA strain isolated from cows with bovine mastitis as well as by other biofilm-forming strains. Through fluorescence microscope observation, LysKΔamidase-treated biofilms were destroyed and became cellular debris.	Not tested	[230]
Csl2	*Streptococcus suis*	Pig (swine industry)	Septicemia, arthritis, endocarditis, pneumonia, and meningitis	New chimeric lysin, Csl2, was developed by fusing the efficient catalytic domain of Cpl-7 and the two CW_7 repeats of the LySMP lysin.	The Csl2 chimera has shown a strong bactericidal activity. The results demonstrated that after the addition of Csl2 to the biofilms formed, the matrix-embedded bacteria were disintegrated, and the number of viable bacteria in the biofilm was reduced by around 2 logs after 1 h of incubation. These results revealed a clear antibiofilm activity of Csl2 on *S. suis*.	Zebrafish death provoked by the *S. suis* (4.5 × 10^7^ CFU per zebrafish) occurred within the first day, and Csl2 injection 1 h after the bacterial challenge reduced the lethality resulting from such infection in a dose-dependent manner. Twenty-four hours after infection, in the group treated with 2 mg/kg of Csl2, the colony forming unit (CFU) numbers in blood have been reduced to 43 CFU per zebrafish. Their results revealed that Csl2 can effectively control bacterial growth in the blood.	[231]
vB_EcoM-UFV13 (UFV13)	*Trueperella pyrogenes*	“Agribusiness Interest Microorganisms Collection of Embrapa Dairy Cattle” (Juiz de Fora, Brazil)	NA	Although the exact action mechanism has not been determined, UFV13 genome sequencing has revealed a broad range of virion-associated hydrolases. The authors hypothesized that heterologous phages with a large number of VAPGHs may possess activity by non-specific hydrolase action against unrelated hosts.	Through crystal violet assay, the biofilm of *T. pyrogenes* was significantly reduced by the phage UFV13.	Not tested	[232]
EW2 AB27 TB49 TriM KRA2 G28	*Escherichia coli*	Poultry skin	*E. coli* infection	The study aimed to characterize phages and composed a phage cocktail suitable for the prevention of infections with *E. coli*. Six phages were isolated or selected from collections and characterized individually and in combination about host range, stability, reproduction, and efficacy in vitro.	Six phages were studied, against three bacterial strains. The phages showed lytic activity against ESBL-producing and avian pathogenic *E. coli* isolates. The formation of biofilm of one strain, E28, was completely prevented by the six-phage preparation.	Not tested	[233]
UPF_BP1 UPF_BP2	*Salmonella* Gallinarum	Viscera pools obtained from birds	Fowl typhoid	The lytic activity of two new bacteriophages (*Salmonella* phages UPF_BP1 and UPF_BP2) against 46 *Salmonella* Gallinarum strains with phenotypic characteristics associated with antimicrobial resistance and biofilm formation.	In vitro, a total of 31 *S.* Gallinarum strains were able to produce biofilm at 22 °C and/or 42 °C and almost 78% (24/31) of these strains were susceptible to both bacteriophages, including six strains classified as biofilm producers at both temperatures.	Not tested	[234]
pSp-J and pSp-S	*Staphylococcus pseudintermedius*	Canine isolates	NA	This study isolated two novel bacteriophages, pSp-J and pSp-S, from canine pet parks in South Korea to potentially control *S. pseudintermedius*.	At lower concentrations, both phages demonstrated a significant effect in preventing the total biomass of the biofilm. The phages were incubated to treat 24 h old *S. pseudintermedius* biofilms, during 24 h without shaking. The results revealed a decrease in the total biomass of biofilm.	Not tested	[235]
Phage phiIPLA-RODI and lytic protein CHAPSH3b	*Staphylococccus aureus*	*S. aureus* isolates from bovine subclinical mastitis	NA	This study aimed to assess the potential interactions between phage phiIPLA-RODI and the phage-derived chimeric lytic protein CHAPSH3b when used together for biofilm removal.	CHAPSH3b inhibits *S. aureus* biofilm formation, presumably by the downregulation of autolysin-encoding genes. The phage vB_SauM_phiIPLA-RODI (phiIPLA-RODI) is also effective in eliminating staphylococcal biofilms. The combination of CHAPSH3b phage with the protein at different concentrations revealed a significant reduction in *S. aureus* biofilm, indicating that there was a synergistic effect in both cases. Through total biomass analysis, the phage significantly reduced the biofilm.	Not tested	[236]
vB_SenM2 vB_Sen-TO17	*Salmonella enterica*	The National Salmonella Centre at the Medical University of Gdansk, Poland	Poultry infection	This study aimed to test the efficacy and safety of two bacteriophages in both in vitro and *Galleria mellonella* in vivo model.	Bacteriophages were either as efficient as antibiotics in reducing the number of living *S. enterica* cells in the biofilm or even more efficient in the case of some strains. In multispecies biofilms (built by *S. enterica*, *P. vulgaris*, *C. freundii*, *E. coli*, and *L. acidophilus*), the effects of phages and antibiotics on biofilms were effective in reducing the number of living *S. enterica* cells.	Treatment with vB_Sen-TO17 led to a significant increase in the survival rate at all tested multiplicities of infection values. vB_SenM-2 treatment was less effective and resulted in an evidently increased survival rate only when phage was used at multiplicity of infection of 100. The highest increase in the survival rate of larvae was observed in the group of animals treated with the cocktail of vB_Sen-TO17 and vB_SenM-2 at all tested multiplicities of infection.	[237]
LP31	*Salmonella* Enteritidis *Salmonella* Pullorum	NA	*Salmonella* spreading in the poultry and food processing industries	In this study, a lytic *Salmonella* phage from poultry feces was isolated and its potential was evaluated for *Salmonella* prevention and control.	In vitro, biofilms of *S. enteritidis* and *S.* Pullorum formed in glass tubes were almost completely removed when phage LP31 was applied.	The phage was introduced in chicks drinking water. The results showed that the concentration of *S. enteritidis* in the feces of chicks whose drinking water contained LP31 was significantly lower than those in the control group. LP31 can be used to effectively reduce the concentration of *S. enteritidis* in the feces of chicks and in the surrounding environment.	[238]
NC5	*Streptococcus uberis*	Bovine isolates	Bovine mastitis	This study describes an optimized engineered endolysin with activity in raw cow’s milk against the bovine streptococcal mastitis pathogen *S. uberis*, by engineering wild-type endolysins with known activity against these targeted pathogens.	The in vitro results revealed that the treatment of *S. uberis* biofilm with 1.5 uM NC5 during 2 h 30 min significantly reduces the biofilm mass of around 70%.	Not tested	[239]
EF-N13	*Enterococcus* *faecalis*	Bovine isolates	Bovine mastitis	New phage (EF-N13) was isolated using the multidrug-resistant *E. faecalis* N13 (isolated from mastitic milk) as the host. The phage EF-N13 belongs to the family *Myoviridae*. The genome of EF-N13 lacked bacterial virulence, antibiotic resistance-, and lysogenesis-related genes.	Phage EF-N13 effectively prevented the formation of biofilms of lysable *E. faecalis*. When phage EF-N13 was incubated with the strain for 24 h and 48 h, *E. faecalis* and its biofilm were almost completely eliminated.	The phage EF-N13 was tested in vivo using the mouse model of mastitis. The multidrug-resistant *E. faecalis* N13 was used to construct the model. After the infection of the mice with N13 and treatment with different titers of phage EF-N13 after 2 h, the number of colonies decreased significantly at 24 h. After 48 h of infection, the bacterial loads in the mammary glands of the mice in the different phage-treated groups were still significantly lower than those in the PBS-treated group.	[240]
**Emulsion of medium-chain triglycerides**							
0·125% *v*/*v* medium chain triglyceride (ML:8) emulsion	*Porphyromonas cangingivalis*, *Porphyromonas salivosa*, *Porphyromonas gingivalis*, *Fusobacterium nucleatum*, *Eikenella corrodens*, *Bacteroides fragilis*, *Prevotella intermedia* *Tanerrella forsythesis*	Canine and feline periodontopathogen isolates	Periodontitis	ML:8 emulsion provides an inhibitory effect on the adhesion of bacteria to surfaces when present as a surfactant coating for the prevention of medical-device-related infection.	The 0·125% *v*/*v* ML:8 emulsion displayed significant activity against biofilm forms of the 10 periodontopathogens investigated within 5 to 10 min exposure.	Not tested	[241]
**Essential oils or their components**							
Syzygium aromaticum and cinnamomum zeylanicum essential oils (EOs)	*Staphylococcus aureus*	Isolates recovered from the milk of cows with subclinical mastitis	Mastitis	Although complex, some antimicrobial action mechanisms are widely documented for EOs, such as permeabilization of cell membranes, plasma membrane depolarization, impairment of lipid polymorphism, interaction with the outer membrane proteins in Gram-negative bacteria, affecting the respiratory processes, coagulating the cytoplasmic material, and depletion of intracellular ATP.	The results showed a significant inhibition of biofilm production by *S. aromaticum* EO on polystyrene and stainless steel surfaces (69.4 and 63.6%, respectively). Its major component, eugenol, was less effective on polystyrene and stainless steel (52.8 and 19.6%, respectively). Both *C. zeylanicum* EO and its major component, cinnamaldehyde, significantly reduced biofilm formation on polystyrene (74.7 and 69.6%, respectively) and on stainless steel surfaces (45.3 and 44.9%, respectively). These findings suggest that these EOs may be considered for applications such as sanitization in the food industry.	Not tested	[242]
*Thymus sibthorpii*, *Origanum vulgare*, *Salvia fruticosa*, and *Crithmum maritimum* EOs	*Staphylococcus aureus*	Isolates recovered from the milk of goats	Mastitis	Although complex, some antimicrobial action mechanisms are widely documented for EOs. It is speculated that they disrupt the membrane of bacteria; in particular, they sensitize the phospholipid bilayer of the cell membrane, increasing the permeability and leakage of vital intracellular constituent.	All tested EOs indicated almost 95% inhibition of biofilm formation at their half MIC, while gentamicin sulfate did not show sufficient antibiofilm activity.	Not tested	[243]
Nine EOs prepared from balsam fir (Abies balsamea; branches), cinnamon (Cinnamomum verum; bark), coriander (Coriandrum sativum; seeds), Labrador tea (Ledum groenlandicum; leaves), peppermint (Mentha piperita; fowering herbs), sage (Salvia ofcinalis; flowering tops), sweet marjoram (Origanum majorana; flowering herbs), thyme (Thymus vulgaris; flowering tops), and winter savory (Satureja montana; flowering tops),	*Streptococcus suis*, *Actinobacillus pleuropneumoniae*	Isolates recovered from respiratory infection	Respiratory infection	NA	Treating pre-formed *S. suis* and *A. pleuropneumoniae* biofilms with thyme or winter savory oils significantly decreased biofilm viability. They also observed synergistic growth inhibition of *S. suis* with mixtures of nisin and essential oils from thyme and winter savory. Concentrations of nisin and cinnamon, as well as thyme and winter savory, essential oils that were effective against bacterial pathogens had no effect on the viability of pig tracheal epithelial cells.	Not tested	[244]
Nine commercial EOs, from roman chamomile (*Anthemis nobilis* L.), star anise (*Illicium verum*), lavender (*Lavandula hybrida*), litsea (*Litsea cubeba* (Lour.) Pers.), basil (*Ocimum basilicum* L.), oregano (*Origanum vulgare* L. subsp. *hirticum*), rosemary (*Rosmarinus officinalis* L.), clary sage (*Salvia sclarea* L.), and thyme (*Thymus vulgaris* L.)	*Pseudomonas aeruginosa*, *Staphylococcus aureus*, *Staphylococcus pseudointermedius*, *Aspergillus niger*, *Aspergillus fumigatus*, *Aspergillus terreus*, *Trichosporon* spp., and *Rhodotorula* spp.	Isolates recovered from dogs and cats with otitis externa	Otitis externa	Although complex, some antimicrobial action mechanisms are widely documented for EOs. It is speculated that they disrupt the membrane of bacteria; in particular, they sensitize the phospholipid bilayer of the cell membrane, increasing the permeability and leakage of vital intracellular constituents.	*O. vulgare* and *S. sclarea* showed superior antibacterial activity, even if not against all the strains. *Trichosporon* sp. and *A. terreus* were insensitive to most EOs, while others showed different degrees of sensitivity. In particular, most fungi were inhibited by *O. vulgare* and *R. officinalis*.	Not tested	[245]
EOs’ components—EOCs: geraniol (GE), carvacrol (CA), eugenol (EU), limonene (LI), thymol (TH), and trans-cinnamaldehyde (CIN) (Es)	*Enterococcus* spp., *Staphylococcus* spp., *Pseudomonas* spp.	Strains were isolated throughout meat chain production in a lamb and goat slaughterhouse	Foodborne in meat chain	The combination of EOCs and HLE or EDTA resulted in the inhibitory effect on pathogenic bacteria in the planktonic state as well as developing and established biofilms. Furthermore, HLE and EDTA decrease the gene expression of multidrug EfrAB, NorE, and MexCD efflux pumps as non-specific resistance mechanisms to several antimicrobials. It has also been demonstrated that eugenol treatment decreased the expression of biofilm-related genes (*IcaA*, *IcaD*, and *SarA*). EU inhibited the expression of adhesion genes and the expression of migration-related genes *fliC*, *fimA*, *lpfA*, and *hcpA*, which encode flagellin A and type 1 fimbriae.	The combination of EOCs with HLE or EDTA (disinfectants) showed particularly positive results given the effective inhibition of biofilm formation. The synergistic combinations of EU and HLE/EDTA, TH, CA, GE, LI, or CIN + EDTA/HLE caused log reductions in established biofilms of several strains (1–6 log10 CFU) depending on the species and the combination used, with *Pseudomonas* sp. Strains being the most susceptible.	Not tested	[246]
*Satureja hortensis* EO	*Escherichia coli* and *Salmonella*	Strains isolated from poultry infections	Poultry infections	The antibiofilm properties of the plant essential oils may be due to anti-adhesive activity of the essential oil compounds, inhibition of structure formation such as reduction in exopolysaccharide production, or altering the biofilm-related gene expression.	Regarding antibiofilm activity, the MIC/2 concentration of *S. hortensis* significantly inhibited biofilm formation of *E. coli*. Inhibition of biofilm formation of *Salmonella* was shown at concentration of MIC/2 and MIC/4. As such, *S. hortensis* EO showed the growth inhibition and bactericidal activity against *E. coli* and *Salmonella*.	Not tested	[247]
*Thyme* EO	*Enterococcus faecalis*	Strain isolated from the product of Chinese water-boiled salted duck and naturally resistant to multiple antibiotics	Poultry infections	The cell adherence was reduced, and an inhibition of EPS synthesis in *E. facealis* biofilms occurred.	Thyme EO (128 and 256 mg/mL) significantly inhibited the biofilm formation of *E. faecalis*. Cell adherence and biofilm thickness were decreased in the thyme-EO-treated biofilms.	Not tested	[248]
Basil, cinnamon, clove, peppermint, oregano, rosemary, common thyme, and red thyme EOs	*Streptococcus suis*	Strains isolated from pigs	Pig infections	The EOs of oregano, red thyme, common thyme, and cinnamon showed a notable in vitro bactericidal activity, by vapor and/or direct contact.	The EOs with the major potential in the disk diffusion method were red thyme, common thyme, oregano, and cinnamon, whereas cinnamon did not show vapor activity. In the microdilution test, all the EOs showed notable antimicrobial activity and strong bactericidal power.	Not tested	[249]
90% of pure oregano, thymol, carvacrol	*Salmonella Escherichia coli*	Strains isolated from pig feces	Pig infections	Carvacrol and thymol are the main components of oregano and thyme oils. They have a very similar chemical structure consisting of a system of hydroxyl groups on the phenolic ring, which are required to elicit strong antimicrobial activity.	*E. coli* and *Salmonella* bacteria colony surfaces were thick smooth surfaces in control. However, colony surfaces in blended- and single-essential-oil treatments have shown cracked surface layer compared with colony surfaces in control.	Not tested	[250]
*Thymus vulgaris* L. EO	*Salmonella* spp.	Strains isolated from wild reptiles housed in a Zoo	Wild reptiles	The complex composition of EOs suggests that multiple mechanisms, probably acting synergistically, are involved in their biological effects. These include the ability to alter the structure of the cytoplasmic membrane to have increased permeability, or to increase oxidative stress within microbial cells, leading to their death, in addition to having a potential inhibitory effect on intercellular communication systems (quorum sensing) or the transcription of genes responsible for biofilm production.	All the isolates were also tested with aqueous solutions of EO at different dilutions (5% to 0.039%). Interestingly, EO proved effective both in inhibiting bacterial growth at low dilutions, with MIC and MBC values ranging between 0.078% and 0.312%, and in inhibiting biofilm production, with values ranging from 0.039% to 0.156%.	Not tested	[251]

Aptamers have proven their ability to inhibit and/or reduce the formation of biofilms of various animal pathogens, including *P. aeruginosa* [214,215,216,217,218], *Salmonella* spp. [198,219,221], *S. aureus* [222,223,224], *E. coli* [224,225], and *Streptococcus* spp. [226,227], among others. In the studies presented, the functionality of aptamers can be divided into (1) the drug-delivery system, in which it is coupled to another active compound that has antimicrobial properties, and/or (2) the drug, which is the aptamer itself that has antibiofilm activity [252]. Some have also shown that aptamers simultaneously have these two functions in controlling the formation of biofilms [223]. The antibiofilm functionality of aptamers appears to be associated with the ability to restrict motility and the initial fixation of biofilm formation [219,220,228]. Most of the aptamers selected for biofilm-forming microorganisms appear to have an affinity for flagella, resulting in the restriction of bacterial rotational frequency, due to the increase in the electrostatic repulsion of cells and surfaces [253]. In addition, flagella have a critical mechanosensory role in surface sensing and the initial stages of surface adhesion that leads to the formation of a biofilm, which also makes them more susceptible to the action of antibiotics [253,254]. The conjugation of aptamers targeting a specific biofilm-forming microorganism and antibiotics can thus enhance the antimicrobial effect of both classes of agents (synergistic effect) [219,220]. Although in vivo applicability has not been confirmed in the control of biofilm-forming microorganism infections, in vitro functionality has been demonstrated and they therefore represent a promising alternative in the treatment/control of the animal diseases described above. Whether as drug carriers or antimicrobial agents, aptamers have proven to be flexible and powerful tools.

### 5.2. Bacteriophages

Bacteriophages (phages) and their derived proteins have been proposed as an alternative or complementary strategy to conventional therapeutics, which can help to control the spread of antibiotic resistance in bacterial pathogens [255,256]. One of the remarkable characteristics of phages is their strain specificity when targeting bacteria. This means that phages are recognized for infecting particular strains or types of bacteria, often determined by surface receptors on the bacterial cell wall that the phage can recognize and bind to. This specificity is essential for the efficacy of phage therapy. By targeting specific strains of bacteria, phages can potentially reduce the risk of harming beneficial bacteria in the body [257].

Bacteriophages are non-toxic, effective, and economically viable and have already been used therapeutically [258]. Moreover, it should be considered that phages are the most abundant biological entities on earth, and they can multiply themselves naturally [256]. Normally, bacteriophages act by degrading the structural peptidoglycan present in the bacterial cell wall using two classes of lytic proteins [259]. The virion-associated peptidoglycan hydrolases degrade peptidoglycan in the initial steps of the infection, and endolysins help release the phage progeny during the late phase of the lytic cycle [260]. Endolysins have gained increasing attention due to their rapid activity, their high specificity for the target bacteria, and their efficacy against antibiotic-resistant bacterial strains [261,262]. Lysins hydrolyze the bacterial cell wall by breaking specific bounds of the biotics.

However, some drawbacks need to be considered. One concern is their potential contribution to horizontal gene transfer and the selection of bacteriophage-insensitive mutants during therapy [263]. Moreover, with regard to phage lytic proteins, the concentration of these proteins decreases gradually after administration, and it is important to ensure protein stability under the desired environmental conditions to avoid protein inactivation [257,264].

Several studies have demonstrated that phages can effectively prevent and eliminate biofilm formation. However, only a few studies using phages have been related to the control of biofilm formation in animal-associated infections. Table 2 summarizes bacteriophages with antibiofilm activity in animal diseases.

There are several studies on the application of phages for the treatment of bovine mastitis associated with different pathogens. In 2017, a study conducted by Zhou and colleagues developed an engineered lysin, LysKΔamidase, by fusing the N-terminal 220 amino acids with the C-terminal 105 amino acids of the staphylococcal phage lysin LysK [230]. The authors demonstrated by a live/dead staining kit that LysKΔamidase disrupted biofilms produced by the MRSA strain recovered from cows with bovine mastitis. Fluorescence microscopic investigation showed that LysKΔamidase-treated biofilms were destroyed and turned into cellular debris. In 2023, two studies were developed in the context of bovine mastitis, however, for two different pathogens, *Streptococcus uberis* and *E. faecalis*. Elst et al. described an endolysin with activity against *S. uberis* in raw cow’s milk [239]. After exposure of the *S. uberis* biofilm to a concentration of 1.5 uM NC5 for 2 h 30 min, a significant reduction of around 70% of the biofilm mass was observed. In another study, the phage was isolated using multidrug-resistant *E. faecalis* N13 as a host [240]. The EF-N13 phage was incubated with the strain for 24 h and 48 h, completely eliminating its biofilm. The phage was tested in vivo using a mouse model of mastitis. When the mouse model was treated with different titers of phage EF-N13, the number of colonies decreased significantly after 24 h, maintaining the effect after 48 h of infection.

*S. suis* is an emerging zoonotic pathogen that is capable of causing septicemia, arthritis, endocarditis, pneumonia, and meningitis both in pigs and humans. As such, research has been conducted to apply bacteriophages to control biofilm formation of *S. suis*. In 2017, Vásquez and colleagues constructed a new chimeric lysin, Csl2, to target *S. suis*, which showed strong bactericidal activity [231]. The results demonstrated that after the addition of Csl2 to the biofilms formed, the bacteria embedded in the matrix were disintegrated and the number of viable bacteria in the biofilm was reduced by around 2 logs after 1 h of incubation. These results revealed a clear antibiofilm activity of Csl2 on *S. suis*. Moreover, using the adult Zebrafish, it was shown that the injection of Cls12 reduced the lethality of the infection.

*Salmonella* spp. is a common foodborne pathogen in the poultry industry [237]. Normally, *Salmonella* can be found in the chicken intestine and human salmonellosis is mainly caused by the consumption of contaminated chicken meat [265,266]. In this sense, alternative methods are needed to prevent infectious diseases in poultry farms and to protect poultry-derived foods from bacterial contamination. Several studies have identified and characterized several bacteriophages. In 2020, Rizzo and colleagues isolated two new bacteriophages, UPF_BP1 and UPF_BP2, with lytic activity and showed that both bacteriophages were capable of reducing the biofilm of *S.* Gallinarum isolates [234]. More recently, in 2022, two more bacteriophages, vB_SenM2 and vB_Sen-TO17, were isolated and tested against *S. enterica* [237]. The results demonstrated that both bacteriophages were as effective as antibiotics in reducing the number of live *S. enterica* cells in the biofilm, or more effective for some strains. When the bacteriophages were tested in the *G. mellonella* in vivo model, the highest increase in larval survival was obtained with the cocktail of both bacteriophages [238]. In another study developed by Ge et al., a lytic phage, LP31, was isolated from poultry feces and almost completely removed the biofilms of *S. enteritidis* and *S. pullorum* formed in glass tubes after 1 h of incubation [238]. In addition, when the phage was introduced into the drinking water of chicks, the concentration of *S. enteritidis* in the feces was significantly lower than those in the control group. These results revealed that LP31 can be used to effectively reduce the concentration of *S. enteritidis* in the feces of chicks or even in the surrounding environment [238].

Other studies have been developed to treat infections related to other pathogens, namely *Trueperella pyrogenes* and *E. coli.* In 2018, Duarte and colleagues evaluated the use of vB_EcoM-UFV13 (UFV13) to prevent the biofilm formation of *T. pyrogenes* [232]. Using a crystal violet assay, the UFV13 phage was capable of significantly reducing biofilm formation. More recently, in 2020, Korf and colleagues developed a study to isolate and characterize phages for the prevention of *E. coli* infections suffered by poultry industries [233]. Six phages were investigated, and three bacterial strains were tested. The phages showed lytic activity against ESBL-producing and avian pathogenic *E. coli* isolates. The formation of biofilm of one strain, E28, was completely prevented by the six-phage preparation, revealing that the six phages are promising candidates for in vivo efficacy studies.

Related with companion animals, a recent study performed by Kim and colleagues identified two novel bacteriophages, pSp-J and pSp-S, isolated from canine pet parks in South Korea that potentially have activity against *S. pseudintermedius* [235]. At lower concentrations, both phages demonstrated a significant effect in preventing the total biomass of the biofilm. Moreover, the phages were incubated to treat 24 h old S. *pseudintermedius* biofilms, for 24 h without shaking. The results revealed a decrease in total biofilm biomass.

Although there are only a few studies using bacteriophages to control biofilm formation in animal diseases, these studies reinforce the concept of phages as a promising and effective alternative treatment to combat multi-resistant bacterial pathogens.

### 5.3. Emulsion of Medium-Chain Triglycerides

The antimicrobial properties of free fatty acids have been extensively documented in previous studies [267]. The medium-chain triglycerides containing medium-chain fatty acids can destroy the bacterial colony in the intestinal tract by decreasing the pH [268]. Moreover, they can act as an effective antibiotic against the bacterial growth [269]. Sun and colleagues conducted research indicating that caprylic acid (C8), capric acid (C10), and lauric acid (C12) exhibit notable antimicrobial activity, with lauric and caprylic acid demonstrating particular efficacy against Gram-positive and Gram-negative bacteria, respectively [270]. Lecithin has been incorporated into the formulation primarily as an emulsifying agent to facilitate the integration of medium-chain free fatty acids into the emulsion, thereby enhancing their antimicrobial effects. Interestingly, previous research has highlighted lecithin’s inhibitory effect on bacterial adhesion to surfaces when employed as a surfactant coating for preventing medical-device-related infections [271]. Based on these findings, Laverty and colleagues [241] hypothesize that the ML:8 emulsion formulation may also possess this beneficial property as shown in Table 2. They therefore tested the in vitro antimicrobial efficacy of a non-toxic free fatty acid emulsion against clinically relevant canine and feline periodontopathogens. Briefly, the composition of ML:8 consisted of an oil-in-water emulsion. A mixture of free fatty acids, acids such as caprylic and oleic acid, solubilized in water was promoted by the addition of membrane lipids in the form of lecithin. The concentration of the ML:8 emulsion tested showed significant activity against biofilm forms of the 10 periodontal pathogens studied within 5 to 10 min of exposure. However, further in vivo research is needed to investigate whether such a drinking water additive can improve compliance and ease of use, allowing daily administration to help prevent periodontal disease.

### 5.4. Essential Oils or Their Components

Throughout history, medicinal plants have consistently demonstrated their importance as reservoirs of therapeutic molecules [272]. Currently, they continue to be a valuable resource for the discovery of new drugs. It is important to note that about 300 of the 3000 known essential oils (EOs) are used commercially [273]. EOs, which consist of various terpenoid and phenolic compounds isolated from aromatic plants, have received considerable attention in recent decades. Their wide range of medicinal and antimicrobial properties, including potential efficacy against biofilm formation, make them a subject of great interest [274]. As such, in recent years, the scientific community has increasingly directed its efforts toward investigating plant-based compounds as promising reservoirs for antimicrobial agents specifically targeting biofilm-related infections. This trend reflects a growing recognition of the therapeutic potential inherent in natural sources, emphasizing the importance of exploring and harnessing the bioactive molecules present in medicinal plants for the development of novel pharmaceutical interventions. The antibiofilm activity of EOs against different pathogens responsible for animal infections such as mastitis, otitis, and respiratory infections, among others, has been reported as shown in Table 2.

EOs are recommended by some veterinarians to treat otitis externa in pets, based on their experience, but data about their efficacy in the scientific literature are very scarce [245,275]. It is important to bear in mind that according to several studies, the efficacy of the antimicrobial effect of EOs is extremely dependent on the strain [275]. A particular example is the case of *P. aeruginosa*. In a study carried out by Ebani and Mancianti, T. vulgaris EO was shown to be effective against *P. aeruginosa* human clinical multidrug-resistant isolates by inhibiting the in vitro growth of the biofilm [275]. However, it is important to remark that the *T. vulgaris* EOs did not consistently exhibit antibiofilm activity against this bacterial species, as observed in the case of a *P. aeruginosa* strain isolated from a dog with external otitis [245]. Despite being resistant to several antibiotics, commercial EOs, including *T. vulgaris* oil, did not demonstrate efficacy in combating biofilm formation by this particular strain.

EOs are characterized by their hydrophobic nature, a critical property that facilitates their ability to penetrate bacterial cell membranes once attached to the cell surface. This hydrophobic interaction leads to the accumulation of EOs, disrupting the cell membrane structure and inducing unfavorable changes in cell metabolism, ultimately leading to cell death. The inhibitory effect of EOs on bacterial growth is attributed to their multiple mechanisms, collectively referred to as EO versatility, which demonstrates the diverse ways in which EOs adversely affect bacterial cells [276].

In addition, EOs exhibit antibiofilm activity due to the presence of both hydrophobic and hydrophilic components in their composition. The hydrophobic components penetrate the lipid substances of the cell membrane and reduce biofilm formation. At the same time, the hydrophilic components diffuse through the EPS matrix of the biofilm, contributing to an overall reduction in biofilm formation. As such, it has been hypothesized that these two properties of EOs work in tandem to disrupt bacterial cell membranes and reduce biofilm formation, highlighting the diverse and potent antimicrobial properties of Eos [273].

The lack of reported bacterial resistance prescribed to EOs is considered to be the major advantage of these products over other antimicrobial agents. However, studies of EOs in cell cultures show a dose-dependent cytotoxic effect, described as increased apoptosis and cellular necrosis [275].

### 5.5. Prebiotics, Probiotics, and Postbiotics

Prebiotics are essentially dietary components that resist digestion by the host but exert beneficial effects by selectively stimulating the growth and/or activity of certain beneficial bacteria in the gut [277,278]. These components include a range of compounds including oligosaccharides (such as fructooligosaccharide and mannan-oligosaccharide), polysaccharides, natural plant extracts, protein hydrolysates, and polyols. By promoting the proliferation of beneficial gut bacteria, prebiotics can improve immune function, have antiviral properties, and even aid in mineral absorption and metabolic regulation. The incorporation of prebiotics as feed additives gained traction in the late 1980s. Among the various types of prebiotics, multifunctional oligosaccharides and acidifiers have emerged as particularly promising options in animal husbandry [277]. In the poultry industry, prebiotics have been studied to control the presence of *Salmonella*, as a feeding-based strategy. However, to the best of our knowledge, although biofilm formation is one of the proposed strategies responsible for *Salmonella* persistence and spread on farms, there is a lack of studies testing prebiotics to treat *Salmonella*-associated biofilms [279]. Similarly, in other animal-associated infections caused by biofilms, there are only a few studies evaluating prebiotics to treat biofilms.

Probiotics are live microorganisms that, when administered in adequate amounts, confer a health benefit to the host, and can be an alternative therapy for controlling biofilm formation. The term ‘probiotics’ was defined by the Food and Agriculture Organization of the United Nations (FAO) and the World Health Organization (WHO) [280].

Probiotic food supplements have been shown to provide health benefits against intestinal infections in animals, and they are used to prevent or treat intestinal disorders [281,282]. In the gut, they can help to balance the microorganisms, affecting the population density [282]. Compared to antibiotics, probiotics are an attractive alternative treatment, as antibiotics can have a destructive effect on the gut microbiota and the resistance acquired by microorganisms [281]. The mechanisms of action of probiotics include the inhibition of pathogen adhesion, production of antimicrobial components, competitive exclusion of microorganisms, improvement in intestinal barrier function, reduction in luminal pH, and modulation of the immune system [282,283]. More specifically, it has been described that probiotic supplementation may influence the development of adaptive traits in neutrophils and other innate immune cells [284]. Furthermore, probiotics affect eukaryotic cells through a variety of mechanisms. For example, short-chain fatty acids (SCFAs) can activate specific G-protein-coupled receptors (e.g., GPR41/43) expressed on enteroendocrine L-cells, thereby inducing the secretion of several gut peptides (e.g., GLP-1, GLP-2) involved in the regulation of energy metabolism and gut barrier function. SCFAs can also modulate gene transcription by inhibiting histone deacetylase activity. In addition to SCFAs, some gut microbes interact with host cells through the production of other specific metabolites or cellular components. Such interactions therefore have a variety of effects on the host, ranging from behavioral improvements in psychopathological conditions to effects on skin health and host metabolism through immune interactions and the gut–brain–skin axis. Bacteria that colonize the normal microbiota, such as *Barnesiella*, have also been associated with reduced susceptibility to gut colonization with vancomycin-resistant enterococci, while *Lactobacillus* treatment reduced carriage of multidrug-resistant potential pathogens [285].

The genera commonly used as probiotics in animals are *Lactobacillus* spp., *Bifidobacterium* spp., *Enterococcus* spp., *Pediococcus* spp., and *Bacillus* spp. [281]. Postbiotics are non-viable bacterial products or by-products of probiotics that have biological activity in the host. They have been tested for their potential to combat infections in animals, particularly in livestock farming [212,286].

Considering the ability of pathogens to cause persistent infections through the formation of biofilms, probiotics have been used to prevent or counteract their development [287]. Table 2 describes some of the studies that have reported the use of probiotics against biofilm-associated infections in animals. The pathogens tested against probiotics and postbiotics included *Salmonella* spp. [204,209,210], *S. aureus* [205,207], *E. coli* [208,211], *Campylobacter* spp. [206], *Streptococcus* spp. [213], and *Enterococcus faecium* and *P. aeruginosa* [212].

*Campylobacter jejuni* is the most common cause of foodborne bacterial infections. It is usually found in animals as it is a common avian commensal. In 2021, Erega and colleagues reported that *Bacillus subtilis* PS-216 was able to prevent the formation of *Campilobacter jejuni* biofilm, dispersing the pre-established biofilm and reducing its growth [206]. These results suggest that *B. subtilis* PS-216 is an effective strategy to reduce the transmission of *C. jejuni* and to prevent or reduce its presence in animal husbandry and food processing [206].

*Salmonella* spp. are also responsible for causing foodborne infections. Therefore, several studies have been carried out to control or prevent salmonellosis. According to Tazehabadi and colleagues, *B. subtilis* KATMIRA 1933 and *B. subtilis* amyloliquefaciens B-1895 were able to reduce the biofilm formed by three *Salmonella* strains by about 50% and 30%, respectively [210]. In 2019, Shi et al. demonstrated that *L. reuteri* S5 has a high capacity to inhibit the biofilm of *Salmonella* Enteritidis, as well as to inhibit the synthesis of intracellular proteins and induce damage to the cellular structure, thus preventing bacterial growth [209]. In the same year, Merino et al. reported that *L. kefiri* 8321, *L. kefiri* 83113, and *L. plantarum* 83114, extracted from kefir grains, showed excellent results in preventing the formation of *Salmonella* biofilms. In addition, surface proteins extracted from the three *Lactobacillus* strains were also able to reduce biofilm formation [204].

In a study carried out by Apiwatsiri and coworkers, *L. plantarum* 22 F and 25F and *P. acidilactici* 72 N strains were tested against colistin-resistant *E. coli*. *E. coli* carry the *mcr-1* gene, which confers resistance to colistin, an antibiotic used as a last resort in the treatment of infections caused by multidrug-resistant bacteria [211,288]. These bacteria have been isolated from pigs, chickens, and humans. If left untreated, colistin-resistant *E. coli* spreads very quickly. The three probiotic strains have been shown to remove the biofilm formed by *E. coli* and also limit the spread of antibiotic resistance genes [211]. Oliveira and colleagues also tested three *Lactobacillus* strains against *E. coli*, specifically the enterotoxigenic *Escherichia coli* (ETEC). This bacterium is the main cause of neonatal diarrhea in pig production. *L. plantarum* 7.1 proved to be the most effective in controlling the biofilm formed by ETEC in vitro [208].

Other studies have been conducted on the treatment of ruminant infections, such as bovine mastitis. Two studies attempted to control mastitis in cows caused by various pathogens using postbiotics secreted by *Lactobacillus sakei* EIR/CM-1 [216], and by *Lentilactobacillus kefiri* LK1 and *E. faecium* EFM 2 [212]. In the study conducted by Sevin and colleagues, they reported that the co-incubation of postbiotics secreted by *L. sakei* EIR/CM-1 with methicillin-resistant *S. aureus* (MRSA), *S. agalactiae*, and *S. dysgalactiae* subsp. dysgalactiae at concentrations above 7.5 mg/mL reduced biofilm formation by approximately 85% [213]. In another study in 2024, Kim and colleagues found that postbiotics derived from *L. kefiri* LK1 and *E. faecium* EFM 2, isolated from kefir and raw milk, significantly inhibited biofilm formation by strains of *S. aureus*, *E. faecalis*, *P. aeruginosa*, and *E. coli*. Furthermore, the use of postbiotics to combat bovine mastitis has been investigated. In 2023, Sabino and colleagues tested exopolysaccharides produced by *Bacillus* spp. (*Bacillus* spp. 18, *B. altitudinis* 27, and *B. velezensis* TR47II) against several strains of *S. aureus* [207]. They showed that components secreted by *Bacillus* spp. reduced biofilm formation by about 40% in most strains of *S. aureus* without affecting bacterial growth. Additional studies have reported the activity of postbiotics against *S. aureus* that cause foodborne infections. It was shown that a cell-free supernatant of *L. acidophilus* LA5 and *L. casei* 431 can effectively remove *S. aureus* biofilm on different surfaces, such as glass and polystyrene [205].

All the probiotics and postbiotics tested in the various studies were found to be able to inhibit biofilm formation by several pathogens. The mechanisms of biofilm inhibition by probiotics have not yet been fully documented; however, different studies have described the inhibition mechanisms based on their results (Table 2). Further in vitro and in vivo studies are needed to fully understand the mechanisms involved, in order to apply probiotics and postbiotics in the treatment and prevention of biofilm-associated infections in animals.

## 6. Conclusions

Biofilms, as resilient communities of microorganisms, pose a continuing challenge by causing infections and complicating treatment strategies. Recognizing the importance of mitigating biofilm formation is crucial across various animal settings including the farms, the wild, and the companion animals. Emerging approaches to mitigate biofilm formation in animals represent a promising frontier in the fight against infections. These approaches offer new ways to improve treatment outcomes and address the growing threat of antimicrobial resistance in both human and veterinary medicine, which need to be continually explored. However, considering the factors mentioned above, the initial step would be to identify the causes of infection through isolation and typing of the pathogen, followed by testing its susceptibility to antibiotics in order to determine the most effective treatment.

## Figures and Tables

**Figure 1 pathogens-13-00320-f001:**
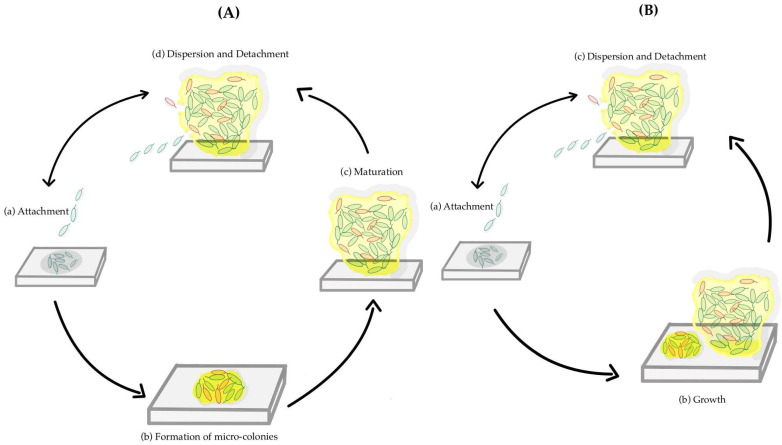
A schematic representation of two models of the formation of a biofilm. The different steps of biofilm formation of the old model or 5-step model (**A**): (a) Attachment; (b) Formation of micro-colonies; (c) Maturation; (d) Dispersion and Detachment. The different steps of a new model or inclusive model (**B**): (a) Attachment; (b) Growth; (c) Dispersion and Detachment.

**Figure 2 pathogens-13-00320-f002:**
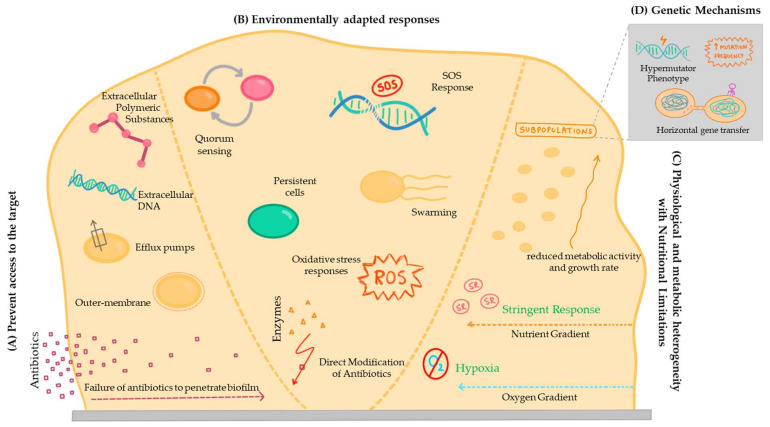
A scheme of the different mechanisms associated with biofilm antimicrobial resistance. (**A**) Prevent access to the target; (**B**) Environmentally adapted responses; (**C**) Physiological and metabolic heterogeneity with nutritional limitations; and (**D**) Genetic mechanisms.

**Table 1 pathogens-13-00320-t001:** Biofilm-associated infections of several animal classes (companion animals; livestock; husbandry; farm; wild animals) and their major etiological agents.

Disorder/Infection	Microorganisms Associated with Biofilm	Animals	References
**Companion Animals**			
**Gastrointestinal System**			
Periodontitis	*Staphylococcus aureus* *Streptococcus pyogenes* *Enterococcus faecalis*	Dogs	[30]
**Auditory System**			
Canine Otitis Externa (OE)	*Staphylococcus pseudintermedius* *Pseudomonas aeruginosa*	Dogs	[3,24,31,32,33]
**Urogenital System**			
Urinary Tract Infections (UTIs)	*Escherichia coli* *Staphylococcus felis* *Pseudomonas aeruginosa*	Dogs, cats	[24,25,32,34,35]
Pyometra (Uterus)	*Escherichia coli* *Staphylococcus* spp. *Streptococcus* spp. *Pseudomonas* spp. *Proteus* spp. *Enterobacter* spp. *Nocardia* spp. *Pasteurella* spp. *Klebsiella* spp.	Female dogs and cats	[3,36,37,38]
**Integumentary System**			
Pyoderma (Skin Infection)	*Staphylococcus pseudintermedius Staphylococcus aureus* *Staphylococcus coagulans Pseudomonas aeruginosa*	Dogs	[21,33,39,40]
Pyoderma (Skin Infection)	*Staphylococcus pseudintermedius* *Staphylococcus aureus* *coagulase-negative staphylococci*	Cats	[21]
Dermatitis	*Streptococcus canis*	Dogs	[41]
Wound Infections (Chronic Nonhealing Pressure Wounds)	*Staphylococcus intermedius* *Staphylococcus epidermidis* *Streptococcus canis*	Dogs	[3,42]
Wound Infections (Postoperative Surgical Site Infection)	*Porphyromonadaceae* *Deinococcaceae* *Methylococcaceae* *Nocardiaceae* *Alteromonadaceae Propionibacteriaceae*	Dogs	[3,43]
Wound Infections (Surgical Suture Segments)	*Staphylococcus* spp. *Streptococcus* spp.	Dogs	[3,44]
**Respiratory System**			
Kennel cough (Infectious Tracheobronchitis)	*Bordetella bronchiseptica*	Dogs	[3]
Nosocomial Infections	*Pseudomonas aeruginosa*	Dogs and cats	[45]
Pneumonia	*Klebsiella* spp.	Dogs and cats	[46]
**Livestock/Husbandry/Farm Animals**			
**Systemic**			
Glaser’s Disease (Polyarthritis, Fibrinous Polyserositis, Meningitis)	*Haemophilus parasuis*	Pigs	[22]
Hemorrhagic Septicemia	*Pasteurella multocida serogroup B:2*	Buffalo, cattle	[47]
Avian Colibacillosis (Airsacculitis, Pericarditis, Peritonitis, Salpingitis, Polyserositis, Colisepticemia, Diarrhea, Synovitis, Osteomyelitis, and Swollen Head Syndrome)	Avian pathogenic *Escherichia coli* (APEC)	Poultry	[48,49,50]
Toxic Shock Syndrome	*Staphylococcus aureus*	Farm horses	[51]
**Respiratory System**			
Porcine Respiratory Disease Complex	*Actinobacillus pleuropneumoniae* *Streptococcus suis* *Pasteurella multocida* *Bordetella bronchiseptica*, *Haemophilus pasasuis* *Mycoplasma hyopneumoniae*	Pigs	[22,23]
Porcine Atrophic Rhinitis	*Bordetella bronchiseptica* *Pasteurella multocida*	Pigs	[52]
Bovine Respiratory Disease Complex (BRDC) or Shipping Fever Pneumonia	*Histophilus somni*	Calves	[3,53]
**Skeletal System**			
Arthritis	*Streptococcus suis* *Haemophilus parasuis*	Pigs	[22]
Osteomyelitis	*Staphylococcus* spp. *Streptococcus* *Escherichia coli* Other Gram-negative bacteria *Erysipelothrix rhusiopathiae* (swine) *Trueperella pyogenes* (cattle)	Horses, swine, broilers, turkeys	[3]
**Cardiovascular System**			
Bovine Myocarditis	*Histophilus somni*	Bovine	[3,53]
**Reproductive System**			
Bovine Mastitis	*Staphylococcus aureus* *Staphylococcus haemolyticus Pseudomonas aeruginosa* *Trueperella pyogenes* *Streptococcus agalactiae* *Staphylococcus epidermidis* *Klebsiella pneumoniae*	Cows, buffalo	[24,35,54,55,56,57,58,59,60,61,62,63,64,65,66,67,68,69,70,71,72,73,74,75]
Goat and Sheep Mastitis	*Staphylococcus aureus*	Goat and sheep	[76,77]
Metritis and Endometritis	*Trueperella pyogenes* *Pseudomonas aeruginosa*	Dairy cattle, horses	[24,35,78,79]
**Nervous System**			
Encephalitis	*Haemophilus parasuis*	Pigs	[22]
Meningitis	*Streptococcus suis* *Haemophilus parasuis*	Pigs	[22,80]
Thrombotic Meningoencephalitis (TME)	*Histophilus somni*	Bovine	[53]
Equine Recurrent Uveitis	*Leptospira* spp.	Equines	[3]
**Gastrointestinal System**			
Poultry Gastroenteritis	*Helicobacter pullorum*	Poultry (broiler chickens)	[27]
Colibacillosis (Camel Calf’s Diarrhea)	Pathogenic *Escherichia coli* F17+	Camel/dromedaries calves (husbandry)	[81,82]
Clostridial Necrotic Enteritis (NE)	*Clostridium perfringens* type A	Poultry	[26]
Diarrhea and Enterotoxaemia	*Clostridium perfringens* type A	Cattle, sheep, goats	[26,83]
Clostridial Enterocolitis	*Clostridium perfringens* type A	Horses, donkeys, foals	[26,84]
Salmonellosis	*Salmonella enterica* *S. Typhimurium* *S. Thompson* *S. Indiana* *S. Pullorum* *S. Heidelberg* *S. Weltevreden* *S. Enteritidis*	Poultry	[85,86,87,88]
Hemorrhagic Colitis	Shiga toxin-producing *Escherichia coli* (STEC)	Livestock	[89]
**Urogenital System**			
Hemolytic Uremic Syndrome	Shiga toxin-producing *Escherichia coli* (STEC)	Livestock	[89]
**Integumentary System**			
Wound Infections	*Pseudomonas aeruginosa*, *Staphylococcus aureus*	Equines	[3,90,91]
**Wild Animals**			
**Gastrointestinal System**			
Hemorrhagic Colitis	Shiga toxin-producing *Escherichia coli* (STEC)	Deer, elk, boar, buffalo, bison, and fox	[89]
Enteric Infection	*Clostridium perfringens*	Free-living rodents and shrews	[83]
Salmonellosis		Exotic pet reptiles	
**Urogenital System**			
Hemolytic Uremic Syndrome	Shiga toxin-producing *Escherichia coli* (STEC)	Deer, elk, boar, buffalo, bison, fox	[89]
**Systemic**			
Hemorrhagic Pneumoniae	*Pseudomonas aeruginosa*	Mink, foxes	[24]

## Data Availability

Not applicable.

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
