# Peer review of "Emerging Approaches for Mitigating Biofilm-Formation-Associated Infections in Farm, Wild, and Companion Animals"

_pathogens, 2024, doi:10.3390/pathogens13040320_

Round 1

Reviewer 1 Report

Comments and Suggestions for Authors

The manuscript entitled “Emerging approaches for mitigating biofilm formation-associated infections in farms, wild and companion animals: a review”, by Araujo et al, focuses on the state-of-art of biofilms-forming bacteria affecting to farm, wild and companion animals, related diseases and impact on animal health and welfare. 

It is a well-written and complete review that provides a wide and comprehensive perspective to understand the microbiological causes of biofilms associated infections in animals and consequently to control and mitigate biofilm formation. 

However, there are some key points to be addressed to improve the quality of the manuscript:

-       Some references could be added in section 1 (Introduction), since it only contains three references. 

-       Figure 1 and 2 should be included in only one figure with two different parts (A: old model or 5-stept model; and B: new model or inclusive model), since both represent two different models of biofilm formation. Furthermore, figure should be improved including more details information of each process represented as well as more information in the figure legend. Finally, as it is explained in the text, the attachment step is a reversible process so could it be included in the figure with a double arrow towards the dispersion and detachment step?

-       In line 111, it is stated that pilli and flagella are required to adhere to the surface but they are appendixes related to movement or exchange of genetic information. What about fimbriae? This information is repeated in line 117, where it is also included fimbriae, which are specifically involved in attachment. 

-       Which are specific gene products expressed during the stage of maturation of the biofilm? Line 137

-       Table 1 and Table 2 should be revised to avoid italics in “spp.”, “other gram-negative bacteria” or “pathogenic”. In table 1, “E. coli” or “T. pyogenes” should be written in the full name. 

-       Some reference is required in section 3.1.3 pyoderma skin infection (lines 250-257).

-       I don´t like Figure 3, it is not very clear, it should be improved to clarify the different mechanisms exposed in the text, for instance, by including the same title of the sections below.  

-       In section 4.1, some specific examples of bacteria and/or antibiotic that use the mechanisms exposed in the text should be included in every sub-section. For instance, in 4.1.1., which antibiotic does it fail to penetrate biofilm or is related to eDNA? Which disease?

-       Which are some examples of specific genes expressed in quorum sensing? Line 693

-       An explanation to clarify which is (p)ppGpp should be included in line 732. 

-       Table 2 should be mentioned in the text in section 5, line 782

-       Phages are recognized to be strain-specific when infecting a bacterium, but it is not included in the text. 

-       Which is the mechanism of action of emulsion of medium chain triglycerides as antimicrobial agents? Section 5.3. 

-       Section 5.5 what about prebiotics? Please revise “Postbiotics and posbiotics” along this section. 

Please, check and revise the reference section to correct some mistakes in volume, page number, name of journal (sometimes full, sometimes abbreviated…). For instance, references 2, 66, 121, 126, 

Author Response

Comments and Suggestions for Authors

The manuscript entitled “Emerging approaches for mitigating biofilm formation-associated infections in farms, wild and companion animals: a review”, by Araujo et al, focuses on the state-of-art of biofilms-forming bacteria affecting to farm, wild and companion animals, related diseases and impact on animal health and welfare.

It is a well-written and complete review that provides a wide and comprehensive perspective to understand the microbiological causes of biofilms associated infections in animals and consequently to control and mitigate biofilm formation.

Authors answer: We appreciate the reviewer's feedback, and we acknowledge the reviewer has carefully analyzed the manuscript.

 However, there are some key points to be addressed to improve the quality of the manuscript:

  1. “Some references could be added in section 1 (Introduction), since it only contains three references.”

Authors answer: Thank you for your comment. Additional references were added to the introduction section.

  1. “Figure 1 and 2 should be included in only one figure with two different parts (A: old model or 5-stept model; and B: new model or inclusive model), since both represent two different models of biofilm formation. Furthermore, figure should be improved including more details information of each process represented as well as more information in the figure legend. Finally, as it is explained in the text, the attachment step is a reversible process so could it be included in the figure with a double arrow towards the dispersion and detachment step?”

Authors answer: We appreciate the reviewer’s comment. As such, we have included both figures in only one figure. Additional information was added to the figure legend in the revised version of the manuscript. Moreover, in both figures, it was included a double arrow between the attachment step and the dispersion and detachment step (see lines 100-104).

  1. “In line 111, it is stated that pilli and flagella are required to adhere to the surface but they are appendixes related to movement or exchange of genetic information. What about fimbriae? This information is repeated in line 117, where it is also included fimbriae, which are specifically involved in attachment.”

Authors answer: We appreciate the reviewer’s correction. Information about fimbriae was lacking in the manuscript and as such additional information was added in lines 116-120.

  1. “Which are specific gene products expressed during the stage of maturation of the biofilm? Line 137”

Authors answer: Thank you for your comment. To make this sentence more clear, additional information was added about the role and composition of EPS in biofilm formation (Lines 142-146).

  1. “Table 1 and Table 2 should be revised to avoid italics in “spp.”, “other gram-negative bacteria” or “pathogenic”. In table 1, “E. coli” or “T. pyogenes” should be written in the full name.”

Authors answer: We appreciate the reviewer’s corrections. As such, we have corrected all issues in the revised version of the manuscript.

  1. “Some reference is required in section 3.1.3 pyoderma skin infection (lines 250-257).”

Authors answer: We appreciate the reviewer's comment. Some references are missing in this section on pyoderma and we have added the appropriate references in the revised version of the manuscript (see lines 248-253).

  1. “I don´t like Figure 3, it is not very clear, it should be improved to clarify the different mechanisms exposed in the text, for instance, by including the same title of the sections below.”

Authors answer: Thank you for your comment. Figure 3 has been improved and the same title for the mechanisms of resistance and tolerance has been added as we used in the text (see lines 542-546).

  1. “In section 4.1, some specific examples of bacteria and/or antibiotic that use the mechanisms exposed in the text should be included in every sub-section. For instance, in 4.1.1., which antibiotic does it fail to penetrate biofilm or is related to eDNA? Which disease?”

Authors answer: The reviewer has a valid point to improve the manuscript. As such, we have added some examples of bacteria and antibiotics, along with the mechanisms of resistance/tolerance they exhibit (see below) in the revised version of the manuscript.

Enterococcus faecium

Mechanism: Altered target site

Antibiotic: Vancomycin

Disease: Enterococcal infections, such as urinary tract infections, bloodstream infections, and endocarditis

Explanation: Some strains of E. faecium have acquired mutations in the cell wall precursors to which vancomycin binds. These alterations in the target site reduce the affinity of vancomycin for its target, thereby limiting its effectiveness in inhibiting cell wall synthesis.

Escherichia coli

Mechanism: Efflux pump overexpression

Antibiotic: Tetracycline

Disease: Urinary tract infections, gastrointestinal infections, bloodstream infections

Explanation: Some strains of E. coli can overexpress efflux pumps, which are proteins responsible for pumping antibiotics out of the bacterial cell. This efflux mechanism reduces the intracellular concentration of antibiotics like tetracycline, leading to antibiotic resistance.

Klebsiella pneumoniae

Mechanism: Carbapenemase production

Antibiotic: Imipenem

Disease: Pneumonia, bloodstream infections

Explanation: K. pneumoniae can produce carbapenemase enzymes, which hydrolyze carbapenem antibiotics like imipenem. This enzymatic modification of the antibiotic prevents it from exerting its bactericidal effect, contributing to carbapenem resistance in K. pneumoniae infections.

Pseudomonas aeruginosa

Mechanism: Biofilm formation

Antibiotic: Ciprofloxacin

Disease: Pseudomonas infections, such as pneumonia, urinary tract infections, and wound infections

Explanation: P. aeruginosa is known for its ability to form biofilms, which are structured communities of bacteria enclosed in a self-produced matrix. Biofilms can act as a barrier, preventing antibiotics like ciprofloxacin from effectively penetrating and killing the bacteria within.

Pseudomonas aeruginosa

Mechanism: Quorum sensing-mediated virulence and antibiotic resistance

Antibiotic: Ciprofloxacin

Explanation: P. aeruginosa uses quorum sensing to regulate the expression of virulence factors and biofilm formation. High cell density and quorum sensing activation in P. aeruginosa biofilms lead to the upregulation of genes encoding efflux pumps, which expel antibiotics like ciprofloxacin from the bacterial cells, contributing to antibiotic resistance.

Pseudomonas aeruginosa

Mechanism: Impermeable outer membrane

Antibiotic: Polymyxins (e.g., colistin)

Explanation: P. aeruginosa possesses an outer membrane with low permeability, which serves as a barrier against antibiotics like polymyxins. This impermeable outer membrane limits the entry of polymyxins into the bacterial cell, reducing their effectiveness in disrupting the bacterial membrane.

Pseudomonas aeruginosa

Mechanism: Biofilm formation with EPS

Antibiotic: Tobramycin

Explanation: P. aeruginosa is notorious for its ability to form robust biofilms with EPS, which act as a physical barrier that limits the penetration of antibiotics like tobramycin into the deeper layers of the biofilm. This reduces the efficacy of tobramycin in eradicating P. aeruginosa infections.

Staphylococcus aureus

Mechanism: Production of β-lactamases

Antibiotic: Penicillin

Disease: Staphylococcal infections, including skin and soft tissue infections, pneumonia, and bloodstream infections

Explanation: S. aureus can produce β-lactamase enzymes, which degrade β-lactam antibiotics like penicillin. This enzymatic degradation renders the antibiotic ineffective against the bacteria.

Staphylococcus aureus

Mechanism: SOS response-mediated DNA repair

Antibiotic: Methicillin

Explanation: S. aureus can activate the SOS response upon exposure to antibiotics like methicillin, which target cell wall synthesis. The SOS response facilitates DNA repair mechanisms, allowing S. aureus to overcome DNA damage caused by methicillin and survive antibiotic treatment, contributing to the development of methicillin-resistant strains

These examples illustrate how various bacteria employ different mechanisms of resistance or tolerance to antibiotics, leading to treatment challenges in specific diseases.

References to support the comment:

  • Urban-Chmiel R, Marek A, StÄ™pieÅ„-PyÅ›niak D, Wieczorek K, Dec M, Nowaczek A, Osek J. Antibiotic Resistance in Bacteria-A Review. Antibiotics (Basel). 2022 Aug 9;11(8):1079. doi: 10.3390/antibiotics11081079. PMID: 36009947; PMCID: PMC9404765.
  • Christaki E, Marcou M, Tofarides A. Antimicrobial Resistance in Bacteria: Mechanisms, Evolution, and Persistence. J Mol Evol. 2020 Jan;88(1):26-40. doi: 10.1007/s00239-019-09914-3. Epub 2019 Oct 28. PMID: 31659373.

  1. “Which are some examples of specific genes expressed in quorum sensing? Line 693”

Authors answer: We appreciate the reviewer’s comment. Additional information about the specific genes responsible for quorum sensing was added in the revised version of the manuscript (see lines 704-707).

  1. “An explanation to clarify which is (p)ppGpp should be included in line 732.”

Authors answer: We appreciate the reviewer’s comment, and as such, additional information was added to clarify the definition of (p)ppGpp (see lines 755-756).

  1. “Table 2 should be mentioned in the text in section 5, line 782”

Authors answer: Thank you for your comment, and a sentence was added to section 5 (see lines 850-851).

  1. “Phages are recognized to be strain-specific when infecting a bacterium, but it is not included in the text.”

Authors answer: We appreciate the reviewer’s comment. As in section 5.2, we described the role of bacteriophages in combatting biofilm-associated infections, we have added this information there in the revised version of the manuscript, as follows: “One of the remarkable characteristics of phages is their strain specificity when targeting bacteria. This means that phages are recognized for infecting particular strains or types of bacteria, often determined by surface receptors on the bacterial cell wall that the phage can recognize and bind to. This specificity is essential for the efficacy of phage therapy. By targeting specific strains of bacteria, phages can potentially reduce the risk of harming beneficial bacteria in the body.” (see lines 913-919).

References to support the comment:

  • Nobrega, F.L.; Costa, A.R.; Kluskens, L.D.; Azeredo, J. Revisiting Phage Therapy: New Applications for Old Resources. Trends Microbiol 2015, 23, 185–191, doi:10.1016/J.TIM.2015.01.006.

  1. “Which is the mechanism of action of emulsion of medium chain triglycerides as antimicrobial agents? Section 5.3.”

Authors answer: Thank you for your comment. The mechanism of action of medium-chain triglycerides was added in the beginning of the section 5.3 (Lines 1010-1012).

  1. “Section 5.5 what about prebiotics? Please revise “Postbiotics and posbiotics” along this section.”

Authors answer: We appreciate the reviewer for noticing the mistake. As pointed out by the reviewer, we also introduced the prebiotics in the manuscript (see lines 1081-1097).

In conclusion, although prebiotics have an important role in animal husbandry, namely as a feed-based strategy, studies testing the effect of prebiotics on biofilm-associated infections in animals are scarce.

  1. “Please, check and revise the reference section to correct some mistakes in volume, page number, name of journal (sometimes full, sometimes abbreviated…). For instance, references 2, 66, 121, 126,”

Authors answer: We appreciate the reviewer’s comment, and all references were revised in the new version of the manuscript.

Reviewer 2 Report

Comments and Suggestions for Authors

The author's describe the role of biofilm formation in animal diseases trying to understand the multiscade complex process behind, and discussing possible  therapeutic approaches to control biofilm formation. It is a well-written manuscript and from my point of view interesting for the scientific audience. However, I'm missing som updated information about the possibility of these microorganisms to be a crucial determinant of animal health. Lately, there are numerous reports showing that different microbial agents may support increased protection in organisms via a so called process - trained immunity. This process is very interesting since it may support increased protection via innate immune cells, which may be of critical development to be considered here. My suggestions would be to include at least a paragraph discussing this issue! In order to give you some small hints regarding these issue, I would suggest you to read the following papers:

PMID: 36979747 (this review paper discusses different aspects of microbial agents supporting trained immunity or in worst case immune tolerance which should be interesting to be discussed). Another one is also: PMID: 32132681.

- there are some interesing studies showing that S. aureus supports similar responses as other mycotic microbial agents, giving raise to a very unkwnown matter whether gram-positive bacteria may support increased resistance or tolerance (PMID: 35537323, PMID: 32639232),

- similarly gut-microbiota may support atnimicrobial effects by innate immune system in animals (PMID: 35203650).

Moreover, you may find also interesting papers showing that LPS from E.coli supports that induction of increased protection or tolerance depedning on the pathogen dose - this aspect should be discussed too, since altogether with these findings you may cover a very novel aspect and support futher ideas for the community in general.

The aspect of probiotics and their role is not fully understood and I would suggest you to check these papers and enlarge this issue a little bit more. Papers: 1) PMID: 33133082, 2) PMID: 32010640, 3) PMCID: PMC9039956.

Comments on the Quality of English Language

English can be improved and some sentences should be re-formulated in order to give some more flow to the manuscript.

Author Response

Comments and Suggestions for Authors

“The author's describe the role of biofilm formation in animal diseases trying to understand the multiscade complex process behind, and discussing possible therapeutic approaches to control biofilm formation. It is a well-written manuscript and from my point of view interesting for the scientific audience. However, I'm missing some updated information about the possibility of these microorganisms to be a crucial determinant of animal health. Lately, there are numerous reports showing that different microbial agents may support increased protection in organisms via a so called process - trained immunity. This process is very interesting since it may support increased protection via innate immune cells, which may be of critical development to be considered here. My suggestions would be to include at least a paragraph discussing this issue! In order to give you some small hints regarding these issue, I would suggest you to read the following papers: PMID: 36979747 (this review paper discusses different aspects of microbial agents supporting trained immunity or in worst case immune tolerance which should be interesting to be discussed). Another one is also: PMID: 32132681. Some interesting studies are showing that S. aureus supports similar responses as other mycotic microbial agents, giving raise to a very unknown matter whether gram-positive bacteria may support increased resistance or tolerance (PMID: 35537323, PMID: 32639232). Ssimilarly gut-microbiota may support antimicrobial effects by innate immune system in animals (PMID: 35203650).”

Authors answer: We are pleased with the congratulations and for the perception of the relevance of the topic of this review. We apologize for the lack of information about trained immunity. As such, we have added a new section in the mechanisms of tolerance and resistance to describe how the trained immunity works and the studies that described the function of trained immunity as a mechanism of resistance to antimicrobials (see lines 797-827).

  1. “Moreover, you may find also interesting papers showing that LPS from E.coli supports that induction of increased protection or tolerance depending on the pathogen dose - this aspect should be discussed too, since altogether with these findings you may cover a very novel aspect and support futher ideas for the community in general.”

Authors answer: The reviewer has a valid point here. In fact, it has been described that innate immune cells act as the first front line of the immune response to microbial pathogens. To this end, they exhibit an extraordinary sensitivity to recognize and respond to pathogen-associated molecular patterns (PAMPs). Endotoxins like bacterial lipopolysaccharides (LPS) have been shown to induce host responses to as few as 100 invading Gram-negative bacteria, corresponding to femtomoles of LPS (1–3). These highly sensitive reactions set off the efficient elimination of the invading microorganisms (Lajqi et al. 2019). In this sense, recent investigations move long-term adaptive responses of the innate immune system into focus. As such, it has been shown the potential of LPS to prevent animal-associated infections. An example is the udder infections with Escherichia coli, which are a serious problem for the dairy industry. Günther and co-workers showed that a mild transient stimulation of healthy udders with a single low dose of LPS (1 μg/quarter) will not only reduce the severity of a subsequently elicited E. coli mastitis but will protect the udder from colonization with E. coli pathogens for three to ten days. More recently, Lajqi and colleagues, suggest the profound influence of preceding contacts with pathogens on the immune response of microglia. The impact of these interactions—trained immunity or immune tolerance—appears to be shaped by pathogen dose. As such, we have added this information in the revised version of the manuscript (see lines 828-837).

References to support the comment:

  • Günther J, Petzl W, Zerbe H, Schuberth HJ, Koczan D, Goetze L, Seyfert HM. Lipopolysaccharide priming enhances expression of effectors of immune defence while decreasing expression of pro-inflammatory cytokines in mammary epithelia cells from cows. BMC Genomics. 2012 Jan 12;13:17. doi: 10.1186/1471-2164-13-17.
  • Lajqi T, Lang GP, Haas F, Williams DL, Hudalla H, Bauer M, Groth M, Wetzker R, Bauer R. Memory-Like Inflammatory Responses of Microglia to Rising Doses of LPS: Key Role of PI3Kγ. Front Immunol. 2019 Nov 8;10:2492. doi: 10.3389/fimmu.2019.02492. PMID: 31781091; PMCID: PMC6856213.

  1. “The aspect of probiotics and their role is not fully understood and I would suggest you to check these papers and enlarge this issue a little bit more. Papers: 1) PMID: 33133082, 2) PMID: 32010640, 3) PMCID: PMC9039956.”

Authors answer: We appreciate the reviewer's suggestion and apologize for the lack of clarity. We have therefore addressed the studies mentioned in the question and provided further details on the role of probiotics in the revised version of the manuscript (see lines 1110-1125).

Comments on the Quality of English Language

  1. “English can be improved and some sentences should be re-formulated in order to give some more flow to the manuscript.”

Authors answer: We apologize for any grammatical errors. We will double-check all sections of the manuscript accordingly.

Reviewer 3 Report

Comments and Suggestions for Authors

This paper is a comprehensive review on the mechanisms of action and mitigations strategies of biofil in animals. It contains a wide range of information, and it is helpful for readers who want to have an overview on the topic. The paper can be enhanced by reducing redundancy in information, keeping consistent formatting, and focusing on important details on animal-related biofilm cases and mechanisms.

Major comments:

The title can be either "Emerging Approaches for Mitigating Biofilm Formation-associated Infections in Farm, Wild, and Companion Animals" or "A Review on Emerging Approaches for Mitigating Biofilm Formation-associated Infections in Farm, Wild, and Companion Animals".

The species names should be more coherent throughout the paper - for example, E. coli or Escherichia coli.

It is good that Section 3 is comprehensive but it lacks coherence. Each disease should be introduced by symptoms in animals, impact, infectious agents, and recent findings in biofilms.

Some information at the begining of Section 4 is redundant as it was already mentioned at the introduction. The section on Direct modification of antibiotics may be expanded for the reader.

Figures 1, 2: It may be better to have the shapes in darker colors than the arrows.

Table 1: In the Animal column, the terms 'bitches' and 'queens' can be left out. There is inconsistency in the texts within the table (like spaces, capital letters, and species names).

Table 2: Too crowded and wordy - In vitro/in vivo main results may be summarized as bullet points with minimal words? Page 17 is empty.

Comments on the Quality of English Language

Minior comments:

Line 30: Is there any reference to the collaboration of micro-communities?

Line 156: This sentence is incomprehensible "The old model, or 5-steps model is represented by all the steps described above separately, however separates the maturation in maturation I and II." 

Line 190: Tab missing.

Line 214: Full stop missing.

Line 222: What does 'LUT' stand for?

Line 261: Does it mean the experiments were conducted in Portugal? Is it relevant?

Line 271: Other studies or another study

Line 359, 397:  Italics missing.

Line 405: Glässer’s disease should be defined with symptoms, before describing its infectious agents.

Line 629: "Oxidative stress reflects the mechanism that bacteria and biofilms develop" - the wording 'reflects' makes it difficult to understand whether it is an antimicrobial pathway or defensive mechanism.

Line 686: "the mechanism is not yet well understood."?

Line 805, 1078: Full stop missing.

Author Response

This paper is a comprehensive review on the mechanisms of action and mitigations strategies of biofilm in animals. It contains a wide range of information, and it is helpful for readers who want to have an overview on the topic. The paper can be enhanced by reducing redundancy in information, keeping consistent formatting, and focusing on important details on animal-related biofilm cases and mechanisms.

Major comments:

  1. “The title can be either "Emerging Approaches for Mitigating Biofilm Formation-associated Infections in Farm, Wild, and Companion Animals" or "A Review on Emerging Approaches for Mitigating Biofilm Formation-associated Infections in Farm, Wild, and Companion Animals".

Authors answer: We appreciate the reviewer's suggestion and we have changed the title in the revised version of the manuscript.

  1. “The species names should be more coherent throughout the paper - for example, E. coli or Escherichia coli.”

Authors answer: We appreciate the reviewer's comment, so we have only included 'Escherichia coli' in full the first time it appears in the text.

  1. “It is good that Section 3 is comprehensive but it lacks coherence. Each disease should be introduced by symptoms in animals, impact, infectious agents, and recent findings in biofilms.”

Authors answer: We appreciate the reviewer's comment. This review focuses on approaches to mitigate biofilm-associated infection in animals. As such, we have reiterated the essential information on each disease, given the fact that other reviews focus on diseases from a clinical perspective.

  1. “Some information at the begining of Section 4 is redundant as it was already mentioned at the introduction. The section on Direct modification of antibiotics may be expanded for the reader.”

Authors answer: We appreciate the reviewer's comment. Section 4 has been reworded with new information, including examples of bacteria and/or antibiotics that use specific mechanisms of resistance. Of note, it also addresses inquestion#8 of reviewer 1 (please see the answer). 

  1. “Figures 1, 2: It may be better to have the shapes in darker colors than the arrows.”

Authors answer: Thank you for your comment. In the revised version of the manuscript, we have combined Figure 1 and Figure 2 into the same figure. A double arrow has been added between the attachment step and the dispersion step. We have chosen to colour the shapes differently to highlight the cells in the figure.

  1. “Table 1: In the Animal column, the terms 'bitches' and 'queens' can be left out. There is inconsistency in the texts within the table (like spaces, capital letters, and species names).”

Authors answer: We appreciate the reviewer for noticing the mistake. As pointed out by the reviewer, we have replaced these terms with ‘female dogs’ and ‘female cats’ in the revised version of the manuscript.

  1. “Table 2: Too crowded and wordy - In vitro/in vivo main results may be summarized as bullet points with minimal words? Page 17 is empty.”

     Authors answer: We appreciate the reviewer's comments. We have therefore summarized the information in Table 2.

Comments on the Quality of English Language

Minior comments:

  1. “Line 30: Is there any reference to the collaboration of micro-communities?”

Authors answer: We appreciate the reviewer's comment. New references have been added to the Introduction in the revised version of the manuscript.

  1. “Line 156: This sentence is incomprehensible "The old model, or 5-steps model is represented by all the steps described above separately, however separates the maturation in maturation I and II."”

Authors answer: Thank you for your comments. This section has been reworded in the new version of the manuscript. Figure 1 and Figure 2 have been merged and this information has been updated (see lines 100-104).

  1. “Line 190: Tab missing.”

Authors answer: We appreciate the reviewer for noticing the mistake.

  1. “Line 214: Full stop missing.”

Authors answer: We are grateful to the reviewer for pointing out the error, and a period has been added.

  1. “Line 222: What does 'LUT' stand for?”

Authors answer: We are grateful to the reviewer for pointing out this error and have added the definition of LUT to the text (line 219).

  1. “Line 261: Does it mean the experiments were conducted in Portugal? Is it relevant?”

Authors answer: Thank you for your comment. We have removed 'in Portugal' from the sentence.

  1. “Line 271: Other studies or another study”

Authors answer: Thank you for your comment. We have replaced ‘another study’ with ‘other studies’.

  1. Line 359, 397: Italics missing.

Authors answer: We thank the reviewer for pointing out the error. We have added italics to the species.

  1. “Line 405: Glässer’s disease should be defined with symptoms, before describing its infectious agents.”

Authors answer: We appreciate the reviewer's comment. As this review focuses on approaches to mitigate biofilm-associated infections in animals, we have focused on the infectious agents rather than the symptoms in the section on animal diseases.

  1. “Line 629: "Oxidative stress reflects the mechanism that bacteria and biofilms develop" - the wording 'reflects' makes it difficult to understand whether it is an antimicrobial pathway or defensive mechanism.”

Authors answer: We appreciate the reviewer's comment. We have replaced ‘reflects’ with ‘represents’.

  1. “Line 686: "the mechanism is not yet well understood."?

Authors answer: We thank the reviewer for pointing out the error. We have corrected the revised version of the manuscript.

  1. “Line 805, 1078: Full stop missing.”

Authors answer: We appreciate the reviewer for noticing the mistake. A full stop has been added to the sentences (see lines 873 and 1173).

Round 2

Reviewer 2 Report

Comments and Suggestions for Authors

The authors have accordingly responded to my issues raised previously.

Comments on the Quality of English Language

Minor language issues. Can be improved during proof reading step.

Reviewer 3 Report

Comments and Suggestions for Authors

None